# Augmin deficiency in neural stem cells causes p53-dependent apoptosis and aborts brain development

**Ricardo Viais[1], Marcos Fariña-Mosquera[1†], Marina Villamor-Payà[1†], Sadanori Watanabe[2†], Lluís Palenzuela[1], Cristina Lacasa[1], Jens Lüders[1]\***

[1]Institute for Research in Biomedicine (IRB Barcelona), The Barcelona Institute of Science and Technology (BIST), Barcelona, Spain; [2]Division of Biological Science, Graduate School of Science, Nagoya University, Chikusa-ku, Nagoya, Japan

**Abstract** Microtubules that assemble the mitotic spindle are generated by centrosomal nucleation, chromatin-mediated nucleation, and nucleation from the surface of other microtubules mediated by the augmin complex. Impairment of centrosomal nucleation in apical progenitors of the developing mouse brain induces p53-dependent apoptosis and causes non-lethal microcephaly. Whether disruption of non-centrosomal nucleation has similar effects is unclear. Here, we show, using mouse embryos, that conditional knockout of the augmin subunit *Haus6* in apical progenitors led to spindle defects and mitotic delay. This triggered massive apoptosis and abortion of brain development. Co-deletion of *Trp53* rescued cell death, but surviving progenitors failed to organize a pseudostratified epithelium, and brain development still failed. This could be explained by exacerbated mitotic errors and resulting chromosomal defects including increased DNA damage. Thus, in contrast to centrosomes, augmin is crucial for apical progenitor mitosis, and, even in the absence of p53, for progression of brain development.

**\*For correspondence:**
jens.luders@irbbarcelona.org

[†]These authors contributed equally to this work

## Introduction

Spindle assembly crucially depends on microtubule nucleation by the γ-tubulin ring complex (γTuRC). During mitosis, γTuRC generates microtubules through three different pathways: centrosomal nucleation, chromatin-mediated nucleation, and nucleation from the surface of other microtubules (*Meunier and Vernos, 2016*; *Petry, 2016*; *Prosser and Pelletier, 2017*). The latter mechanism is mediated by the augmin complex and has been referred to as a microtubule amplification mechanism (*Goshima et al., 2008*; *Goshima and Kimura, 2010*; *Lawo et al., 2009*; *Uehara et al., 2009*). Augmin binds to the lattice of microtubules generated by the centrosome- and chromatin-dependent pathways and, through recruitment of γTuRC, promotes nucleation of additional microtubules that grow as branches from these sites (*Alfaro-Aco et al., 2020*; *Kamasaki et al., 2013*; *Petry et al., 2013*; *Tariq et al., 2020*). The existence of multiple nucleation pathways may provide some level of redundancy to spindle assembly, but concerted action by multiple nucleation mechanisms has also been described (*Hayward et al., 2014*; *Prosser and Pelletier, 2017*). While functional studies in *Xenopus* egg extract and cultured cell models have generated a wealth of information regarding the types of spindle defects that occur when specific nucleation pathways are compromised, how these defects impinge on cell fate and development remains poorly defined.

Gene mutations that cause functional or numerical centrosome aberrations are associated with primary microcephaly, a developmental disorder that results in the reduced thickness of the cerebral cortex. Depletion of apical progenitors following abnormal mitoses has been identified as a pathogenic mechanism (*Jayaraman et al., 2018*; *Marthiens and Basto, 2020*; *Nano and Basto, 2017*).

Apical progenitors of the developing cerebral cortex are highly polarized cells. Their cell bodies are positioned in the ventricular zone (VZ), while their apical and basal processes contact the ventricular surface (VS) and basal lamina, respectively (*Arai and Taverna, 2017*; *Chenn et al., 1998*; *Chou et al., 2018*). Prior to mitosis, the nucleus migrates apically and mitotic chromosome segregation occurs near the apical surface. Early during cortical development, apical progenitors divide symmetrically, expanding the progenitor pool. At later stages they switch to self-renewing asymmetric mitoses, producing a neuron or intermediate progenitor in each division. Centrosomal microtubules were proposed to be at the core of these fate decisions, by controlling the distribution of cell fate determinants through correct positioning of the mitotic spindle (*Homem et al., 2015*; *Taverna et al., 2014*; *Uzquiano et al., 2018*). Recent work showed that progenitor fate is strongly impacted by mitotic duration. Mitotic delay results in more neurogenic divisions and an increased percentage of progenitors undergoing p53-dependent apoptosis, depleting the progenitor pool (*Mitchell-Dick et al., 2020*; *Pilaz et al., 2016*). Consistently, mitotic delay, premature differentiation, and apoptosis have all been observed for centrosome defects in mouse models of primary microcephaly (*Insolera et al., 2014*; *Lin et al., 2020*; *Marjanović et al., 2015*; *McIntyre et al., 2012*; *Novorol et al., 2013*). Interestingly, in cases where it has been tested, such as *Cenpj-* or *Cep63*-deficient mice, the reduced cortical thickness was fully rescued by co-deletion of *Trp53*, identifying p53-dependent apoptotic cell death as main driver of microcephaly in these models (*Insolera et al., 2014*; *Marjanović et al., 2015*). Recently, it was shown that this response involves the USP28-53BP1-p53-p21-dependent mitotic surveillance pathway, which is triggered by prolonged mitosis resulting from centrosome loss (*Phan et al., 2021*). Depletion of progenitors by apoptosis may be less important in human microcephaly, where organoid models have revealed premature differentiation as the main response (*Gabriel et al., 2016*; *Lancaster et al., 2013*).

The roles of chromatin-mediated nucleation and augmin-dependent amplification in this context are less clear. Mouse embryos deficient for *Tpx2*, a spindle assembly factor that functions in chromatin-mediated nucleation, abort development after a few rounds of highly abnormal mitotic divisions (*Aguirre-Portolés et al., 2012*). Similar observations were made for mouse embryos lacking the expression of the augmin subunit *Haus6* (*Watanabe et al., 2016*). However, since early mouse development occurs in the absence of centrosomes (*Gueth-Hallonet et al., 1993*), the embryos in the above studies lacked two of the three mitotic nucleation pathways.

Early functional studies by augmin knockdown in cell lines described mitotic defects that ranged from relatively mild for *Drosophila* cells (*Goshima et al., 2008*; *Meireles et al., 2009*) to more severe for human cells (*Lawo et al., 2009*), suggesting cell type- or organism-specific differences. Consistent with this, the knockout of augmin in *Aspergillus* has no obvious phenotype (*Edzuka et al., 2014*), *Drosophila* augmin mutants are viable with mild mitotic defects observed in only some cell types (*Meireles et al., 2009*; *Wainman et al., 2009*), and a zebrafish mutant is also viable but displays defects in the expansion and maintenance of the hematopoietic stem cell pool (*Du et al., 2011*). A more recent inducible knockout of the augmin subunit *HAUS8* in non-transformed human RPE1 cells caused mild spindle defects before cells underwent p53-dependent G1 arrest, but co-deletion of *Trp53* exacerbated the mitotic phenotype (*McKinley and Cheeseman, 2017*). This response may involve the USP28-53BP1-p53-p21-dependent mitotic surveillance pathway, which is triggered by centrosome loss or prolonged mitosis (*Fong et al., 2016*; *Lambrus et al., 2016*; *Meitinger et al., 2016*), but this was not directly tested.

To uncover the specific role of augmin-mediated microtubule amplification in mitotic spindle assembly and cell fate determination, we sought to study augmin deficiency in centrosome-containing cells in vivo. To this end, we conditionally knocked out *Haus6* in proliferating apical progenitors in the embryonic mouse brain using nestin promotor-driven Cre expression. We found that augmin is essential for brain development, promoting mitotic progression, and preventing p53-dependent apoptosis in neural progenitors. Intriguingly, while the absence of p53 promoted growth in *Haus6* knockout brains, this was accompanied by exacerbated mitotic errors and disruption of tissue integrity. Our results show that contrary to centrosomal microtubule nucleation, the augmin-dependent pathway is essential for apical progenitor mitotic progression and survival, and thus for brain development.

## Results

### Augmin is essential for proper development of the mouse forebrain

Previous work has shown that the augmin complex is composed of eight subunits and that depletion of any subunit interferes with augmin assembly and function (*Goshima et al., 2008*; *Lawo et al., 2009*; *Uehara et al., 2009*). Mouse embryos that completely lack expression of the augmin subunit *Haus6* do not survive the blastocyst stage (*Watanabe et al., 2016*). In order to test the specific requirement for augmin in proliferating neural progenitors, we obtained floxed *Haus6* mice in which exon 1 of the *Haus6* gene is flanked by loxP sequences (*Watanabe et al., 2016*). To generate *Haus6* conditional knockout (*Haus6* cKO) mice for the current study, we removed the neomycin cassette that was present in the original strain adjacent to exon 1 (see Materials and methods for details). We then crossed these mice with mice expressing Cre recombinase under the control of the Nestin promoter, to induce *Haus6* knockout in apical progenitors starting around day E10.5 (*Figure 1a*; *Figure 1—figure supplement 1a*; *Graus-Porta et al., 2001*; *Tronche et al., 1999*). In contrast to the full knockout (*Watanabe et al., 2016*), *Haus6* cKO mice passed through all developmental stages and at E13.5 we observed efficient deletion of *Haus6* in the brain (*Figure 1—figure supplement 1b*). Whereas mice with a heterozygous *Haus6* deletion developed normally and were fertile, *homozygous Haus6 cKO* mice died around birth. Analysis of *Haus6* cKO animals at E17.5 showed severe defects in brain development, whereas overall body development appeared normal (*Figure 1b,c*; *Figure 1—figure supplement 1c*). Histopathology analysis revealed a strong disruption or absence of different forebrain structures (cortex, thalamus, and hypothalamus) and of the cerebellum (*Figure 1c*; *Figure 1—figure supplement 1c*). To evaluate whether this was due to agenesis or tissue loss during development, we analyzed embryos at E13.5. Even at this earlier stage, brains in *Haus6* cKO embryos displayed severe defects compared to control embryos. Lateral cortexes in *Haus6* cKO embryos were almost completely absent and thalamus structures, while partially formed, displayed a strong reduction in radial thickness (*Figure 1d,f*). Moreover, spaces between tissue structures were filled with cellular debris. These data suggest that, in *Haus6* cKO brains at early developmental stages, formation of structures that would give rise to the cortex, thalamus, and hypothalamus is initiated but not completed, leading to tissue loss and abortion of brain development at later stages.

### Loss of augmin impairs mitotic progression in cortical and thalamic neural progenitors

To analyze defective brain development in *Haus6* cKO animals at E13.5 at the cellular level, we focused on the thalamus, which was at least partially preserved. We co-stained brain sections with antibodies against PAX6 and βIII-tubulin to label apical progenitors and neurons, respectively. In *Haus6* cKO embryos, we observed that the reduced radial thickness in the thalamus was due to a striking thinning of the neuronal layer by ~90% when compared to controls (*Figure 1e,f*), indicating severely impaired neurogenesis. In some parts, where tissue organization appeared to be disrupted, we also observed neurons that were misplaced in apical regions (*Figure 1e*). To directly test if augmin deficiency impaired mitoses, we identified and quantified mitotic cells in the thalamus using Ser10-phospho-Histone H3 (pH3-Ser10) staining. In *Haus6* cKO embryos, we observed a ~4-fold increase in the number of mitotic cells in the region closest to the VS compared to controls, whereas there were no significant differences in more basal regions (*Figure 2a,b*). The percentage of *Haus6* cKO mitotic cells in prometaphase was strongly increased, whereas metaphases and ana/telophases were reduced relative to controls (*Figure 2c*; *Figure 2—figure supplement 1a*). This increase in early and decrease in later mitotic figures were consistent with a delay in spindle assembly. Taken together, these observations suggest that augmin deficiency in progenitors of the thalamus leads to a defect in progression to metaphase, causing mitotic delay.

To analyze cortical progenitors and since there were no intact cortical structures in *Haus6* cKO brains at E13.5, we analyzed embryos at E11.5. At this stage, cortical structures were present suggesting that, as for the thalamus, cortical tissue is originally formed but lost at later stages. Similar to the situation in the thalamus at E13.5, in *Haus6* cKO cortexes at E11.5 the percentage of mitotic progenitors was increased when compared to controls and this occurred specifically in the apical region and not in more basal layers. Again, this increase in mitotic cells was due to accumulation in

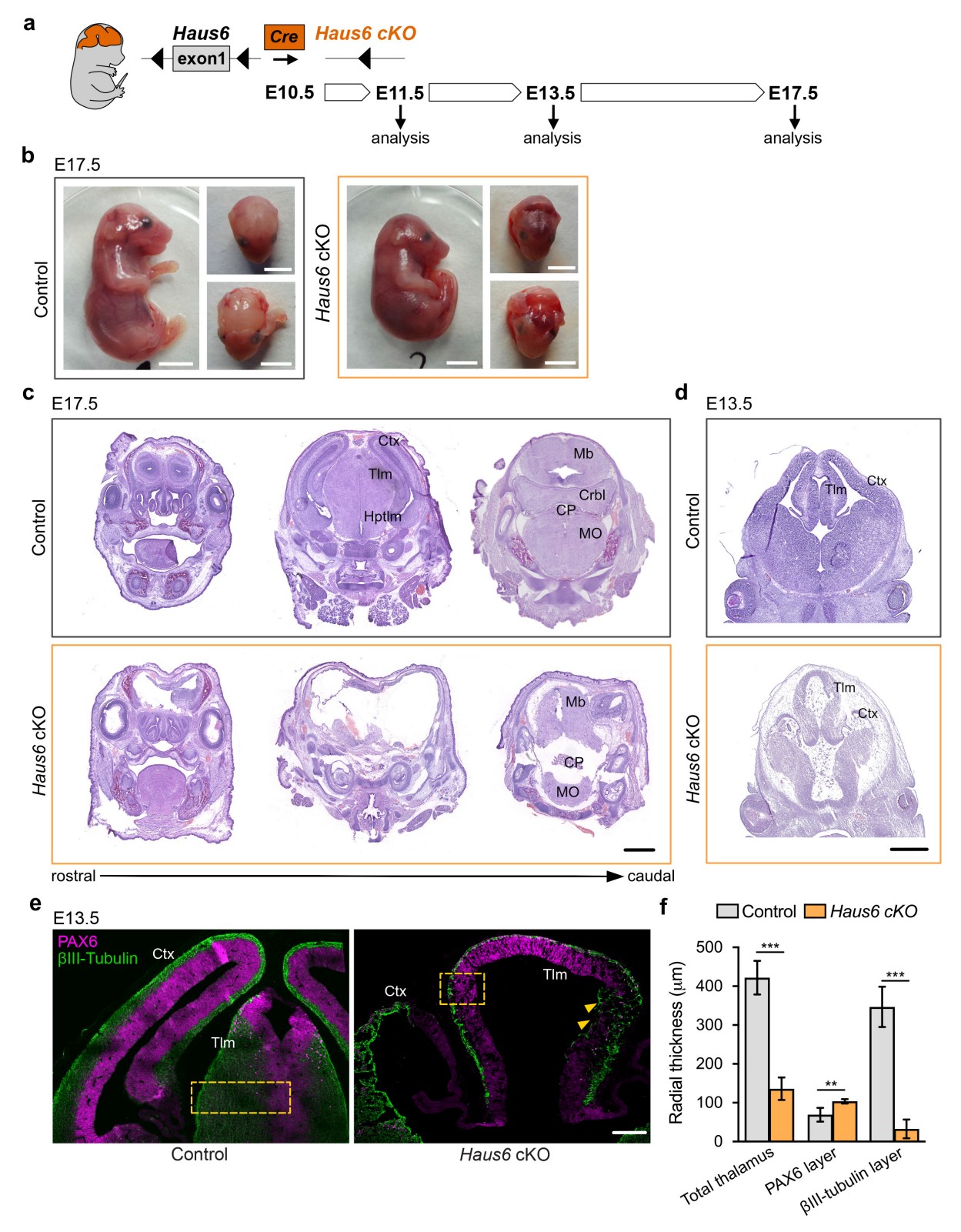

**Figure 1.** Loss of *Haus6* aborts forebrain development. (a) Schematic representation of the experimental strategy used to evaluate the role of augmin during mouse brain development through generation of brain-specific Nestin-Cre *Haus6* cKO embryos. (b) Pictures of E17.5 control (*Haus6*^fl/fl Nestin-Cre⁻) and *Haus6* cKO (*Haus6*^fl/fl Nestin-Cre⁺) embryos. (c, d) Coronal histological sections from (c) E17.5 and (d) E13.5 control and *Haus6* cKO stained with hematoxylin-eosin. Different brain structures are labeled: Ctx (cortex), Tlm (thalamus), Hptlm (hypothalamus), Mb (midbrain), Crbl (cerebellum), MO

*Figure 1 continued on next page*

*Figure 1 continued*

(medulla oblongata), and CP (choroid plexus). (**e**) Representative images of the cortex (Ctx) and thalamus (Tlm) of E13.5 control (*Haus6*<sup>fl/wt</sup> Nestin-Cre[+])
and *Haus6* cKO (*Haus6*<sup>fl/fl</sup> Nestin-Cre[+]) embryos. Coronal sections were stained against PAX6 (magenta – apical progenitors) and βIII-tubulin (green –
neurons). Yellow arrowheads highlight regions of the thalamus where tissue disruption is observed in *Haus6* cKO embryos. Yellow boxes indicate the
regions used for quantifications in (**f**). (**f**) Quantification of the total radial thickness of the thalamus in E13.5 embryos and of layers formed by PAX6- and
βIII-tubulin-positive cells. n=3 for control and n=5 for *Haus6* cKO embryos. Plotted values are means, error bars show SD. **$p<0.01$, ***$p<0.001$ by two-
tailed t-test. Scale bars: (**b**) 5 mm, (**c**) 1 mm, (**d**) 0.5 mm, and (**e**) 150 μm.
The online version of this article includes the following source data and figure supplement(s) for figure 1:

**Source data 1.** Source data associated with *Figure 1f*.
**Figure supplement 1.** Forebrain structures are absent in *Haus6* cKO embryos at E17.5.
**Figure supplement 1—source data 1.** Original agarose gel images associated with *Figure 1—figure supplement 1b*.

---

prometaphase (*Figure 2—figure supplement 1b–e*). Taken together, the data show that augmin
plays an important role in allowing the timely mitotic progression of apical progenitors in different
regions of the developing mouse brain.

To test if augmin-deficient progenitors displayed spindle defects, we analyzed brain sections with
antibodies against γ-tubulin and α-tubulin (*Figure 2d–i*). Mitotic apical progenitors in the thalamus
of control animals displayed strong, centrosomal staining of γ-tubulin at spindle poles and more dif-
fuse γ-tubulin signals along spindle microtubules. In *Haus6* cKO embryos, γ-tubulin could not be
detected on spindle microtubules. Moreover, in ~50% of cells, the staining of γ-tubulin at spindle
poles was dispersed into multiple smaller foci (*Figure 2d,e*). Some of these foci were not associated
with centrioles, as revealed by centrin staining, suggesting that they resulted from PCM fragmenta-
tion rather than centrosome amplification (*Figure 2—figure supplement 1f*). Consistent with this,
centriole numbers in *Haus6* cKO cells were not increased compared to controls (*Figure 2—figure
supplement 1g*). Similar observations were previously made by knockdown of augmin subunits in
cell lines (*Lawo et al., 2009*). Labeling of microtubules by α-tubulin antibodies revealed spindle
abnormalities in about half of the mitotic progenitors in *Haus6* cKO animals (*Figure 2f,g*). This
included cases where spindle microtubules could not be detected (*Figure 2h,i*), suggesting
decreased stability as previously reported (*Goshima et al., 2008*; *Lawo et al., 2009*; *Zhu et al.,
2008*). Defective spindles in *Haus6* cKO cells lacked the bipolar configuration with two robust and
focused microtubule asters typically seen in controls. Instead, spindle microtubules were associated
with multiple, scattered γ-tubulin foci, resulting in spindles that appeared disorganized, sometimes
with multiple poles (*Figure 2f*). However, bipolar configurations including at ana/telophase were
also observed and cell divisions occurred in *Haus6* cKO progenitors, suggesting that mitosis was not
completely blocked.

Considering that augmin-deficiency caused pole fragmentation, we wondered whether this
affected spindle positioning. We measured spindle angles relative to the VS in dividing apical pro-
genitors in the thalamus and in the cortex of E13.5 and E11.5 *Haus6* cKO embryos, respectively. We
found that in both cases the majority of spindles axes were oriented horizontally similar to spindles
in control cells (*Figure 2—figure supplement 2h–j*). This is consistent with results from previous
work showing that the presence of multiple spindle poles in progenitors due to extra centrosomes
does not significantly affect spindle orientation (*Marthiens et al., 2013*).

In summary, augmin deficiency in apical progenitors disrupts the recruitment of γ-tubulin to spin-
dle microtubules, causes pole fragmentation, and interferes with bipolar spindle assembly and
mitotic progression.

## Loss of augmin in neural progenitors induces p53 expression and apoptosis

We sought to determine the fate of progenitors undergoing abnormal mitoses after the loss of aug-
min. We probed thalamus and cortex of E13.5 and E11.5 *Haus6* cKO embryos, respectively, for p53
induction and the presence of the apoptotic marker cleaved caspase-3. Indeed, p53 and cleaved
caspase-3 were strongly upregulated in both brain regions (*Figure 2j,k*; *Figure 2—figure supple-
ment 2a*), whereas cells positive for these markers were barely found in the corresponding tissues of
control embryos. To reveal the identity of cells overexpressing p53, we performed a triple staining
with antibodies against p53, the neuronal marker βIII-tubulin, and the apical progenitor marker

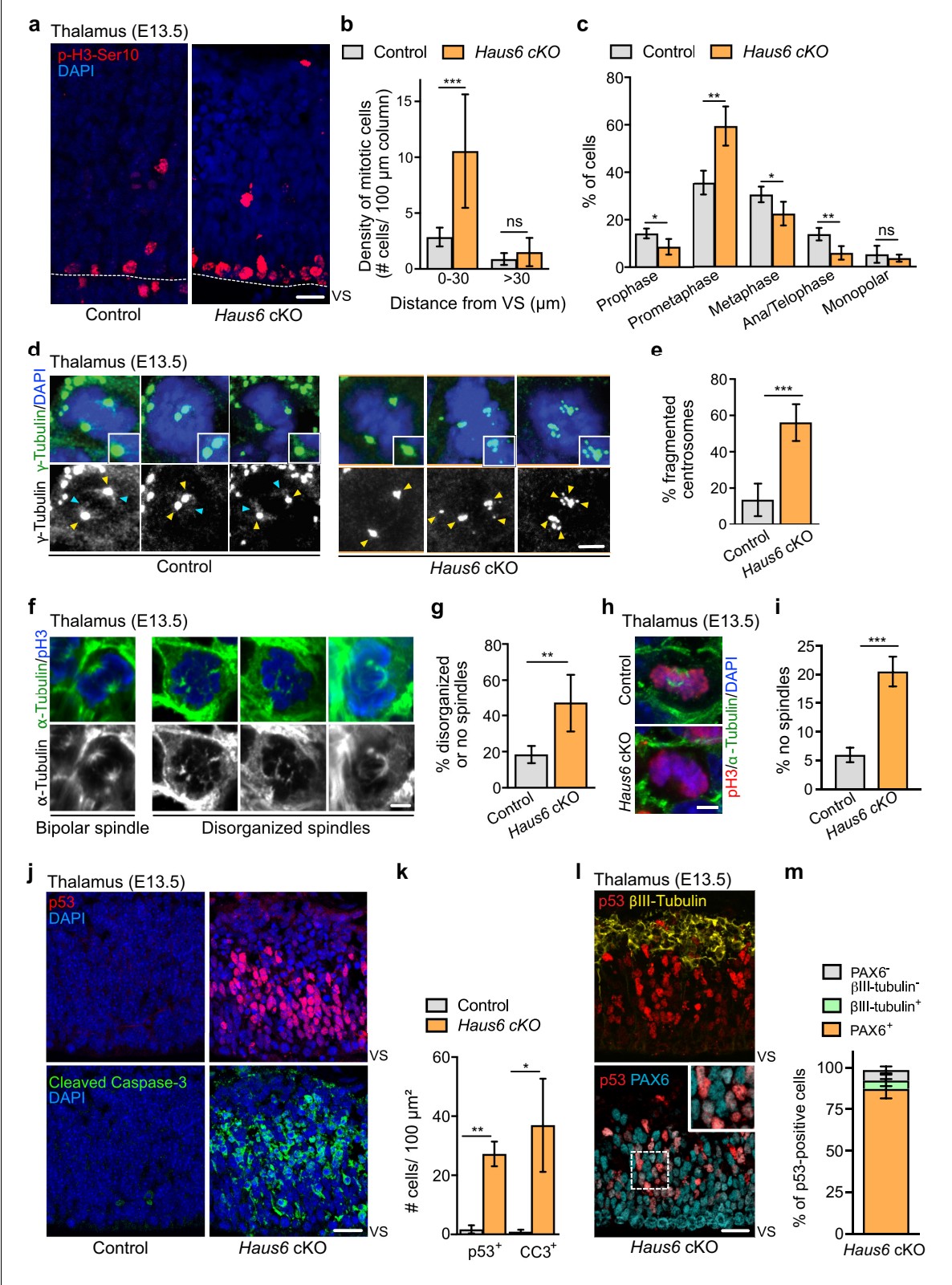

**Figure 2.** Augmin deficiency in neural progenitors impairs mitotic spindle assembly and induces p53 expression and apoptosis. (**a**) Representative images of phospho-Histone H3 (pH3) positive mitotic cells in the thalamus of E13.5 control (*Haus6*fl/wt Nestin-Cre+) and *Haus6* cKO (*Haus6*fl/fl Nestin-Cre+) embryos. Staining of the mitotic marker pH3-Ser10 in red and DAPI in blue. (**b**) Quantification of the density of mitotic cells close to the ventricular surface (VS) (<30 μm away) and in outer layers of the cortical plate (>30 μm away). n=4 for control and n=4 for *Haus6* cKO embryos (total of

*Figure 2 continued on next page*

*Figure 2 continued*

203 and 697 mitotic cells, respectively, in 2–4 sections per embryo). (c) Quantification of mitotic progenitors at different mitotic stages. n=4 for control and n=5 for *Haus6* cKO embryos (total of 261 and 427 mitotic cells, respectively, in one section per embryo). (d) E13.5 control and *Haus6* cKO coronal thalamus sections stained with antibodies against γ-tubulin (green) and DAPI to label DNA (blue). Yellow arrowheads point to γ-tubulin staining at spindle poles. Light blue arrowheads point to spindle-associated γ-tubulin staining. Insets are 1.4× magnifications of spindle poles in the example cells. (e) Quantification of the percentage of mitotic cells in (d) with fragmented centrosomes. n=5 for control and n=3 for *Haus6* cKO embryos (total of 135 and 198 cells counted, respectively, 18–69 cells per embryo). (f) Coronal thalamus sections stained with antibodies against α-tubulin (green) and pH3 (blue). (g) Quantification of the percentage of mitotic progenitors from control and *Haus6* cKO E13.5 embryos displaying disorganized or no spindles. n=5 for control and n=3 for *Haus6* cKO embryos (total of 152 cells and 90 cells counted, respectively, 27–32 cells per embryo). (h) Examples of *Haus6* cKO mitotic progenitors in the E13.5 thalamus in which spindle microtubules, stained with anti-α-tubulin antibodies, cannot be detected, whereas spindles are present in control cells. (i) Quantification of cells as in (h) without detectable spindle microtubules. n=3 for control and n=4 for *Haus6* cKO embryos (total of 216 and 243 cells counted, respectively, 40–88 cells per embryo). (j) Representative images of control (*Haus6*<sup>fl/wt</sup> Nestin-Cre<sup>+</sup>) and *Haus6* cKO (*Haus6*<sup>fl/fl</sup> Nestin-Cre<sup>+</sup>) coronal thalamus sections stained with an antibody against p53 (red – upper panel) and the apoptotic marker cleaved caspase-3 (green – lower panel). DNA is labeled with DAPI (blue). (k) Quantification of the density of p53- and cleaved caspase-3-positive cells in the E13.5 thalamus in brain sections as shown in (j). n=4 for control and n=3 for *Haus6* cKO embryos for quantifications of p53 positive cells and n=3 for control and n=3 for *Haus6* cKO embryos for cleaved caspase-3-positive cells (an area between 460 and 3950 µm² quantified per embryo). (l) Representative images of *Haus6* cKO coronal brain sections showing the thalamus stained for p53 (red), PAX6 (cyan), and βIII-tubulin (yellow). The inset is a 1.8× magnification of cells with nuclei staining positive for both p53 and PAX6. (m) Quantification of the percentage of *Haus6* cKO cells showing induction of p53 and co-expressing PAX6 (orange), βIII-tubulin (light green), or none of these markers (gray). E13.5 thalamus regions, n=3 for different *Haus6* cKO embryos (total of 1020 p53-positive cells). (b, c, e, g, i, k, m) Plotted values are means, error bars show SD. *p<0.05, **p<0.01, ***p<0.001 by two-tailed t-test. Scale bars: (a) 20 µm, (d, f, h) 3 µm, and (j, l) 25 µm.

The online version of this article includes the following source data and figure supplement(s) for figure 2:

**Source data 1.** Source data associated with *Figure 2b, c, e, g, i, k m*.
**Figure supplement 1.** *Haus6* cKO induces mitotic defects in the cortex at E11.5.
**Figure supplement 1—source data 1.** Source data associated with *Figure 2—figure supplement 1d, e, g, i, j*.
**Figure supplement 2.** *Haus6* cKO induces apoptosis and cell cycle arrest.
**Figure supplement 2—source data 1.** Source data associated with *Figure 2—figure supplement 2c*.

PAX6 (*Figure 2l*). This experiment showed that in the *Haus6* cKO thalamus ~87% of the p53-positive cells were also positive for PAX6 and only a minor fraction (~5%) for βIII-tubulin (*Figure 2m*). Moreover, we observed that PAX6-positive progenitors displaying p53 induction were exclusively interphase cells, based on the presence of intact nuclei. We concluded that p53 induction occurred specifically in augmin-deficient progenitors, after exit from abnormal mitoses. Interestingly, some cells in the thalamus of *Haus6* cKO embryos also displayed upregulated expression of the cell cycle inhibitor p21 (*Figure 2—figure supplement 2b,c*).

Taken together, the data suggests that mitotic spindle defects in *Haus6* cKO progenitors are not catastrophic per se, but efficiently trigger cell cycle arrest and apoptotic cell death upon completion of mitosis.

## Co-deletion of *Trp53* in *Haus6* cKO embryos rescues apoptosis but not forebrain development

Since massive apoptosis in *Haus6* cKO brains was correlated with p53 induction, we wondered whether cell death was p53-dependent and the cause of aborted brain development. To address this, we crossed *Haus6* cKO mice with *Trp53* KO mice (*Figure 3a*). Strikingly, at E13.5, a stage at which *Haus6* cKO brains displayed massive apoptosis, lacked cortical structures, and had a poorly developed thalamus, *Haus6* cKO *Trp53* KO brains showed only minimal signs of apoptosis and there was some growth in the regions where cortex and thalamus would be expected to form (*Figure 3b–d*). Consistent with this, there was also no upregulation of p21 (*Figure 3—figure supplement 1a–d*). Tissue growth was enhanced when compared to the single *Haus6* cKO brains, but seemed to lack the layered organization observed in control brains at this stage (*Figure 3b*). At E17.5, however, when thalamus and cortex were well formed in controls, in *Haus6* cKO *Trp53* KO embryos cortex and thalamus structures appeared thin and undeveloped (*Figure 3e*). Moreover, as observed for *Haus6* cKO embryos, *Haus6* cKO *Trp53* KO animals were not viable and died around birth.

In summary, massive apoptosis and cell cycle arrest in *Haus6* cKO brains are rescued in *Haus6* cKO *Trp53* KO brains, promoting growth in the affected brain regions, but this growth is not productive for proper brain development.

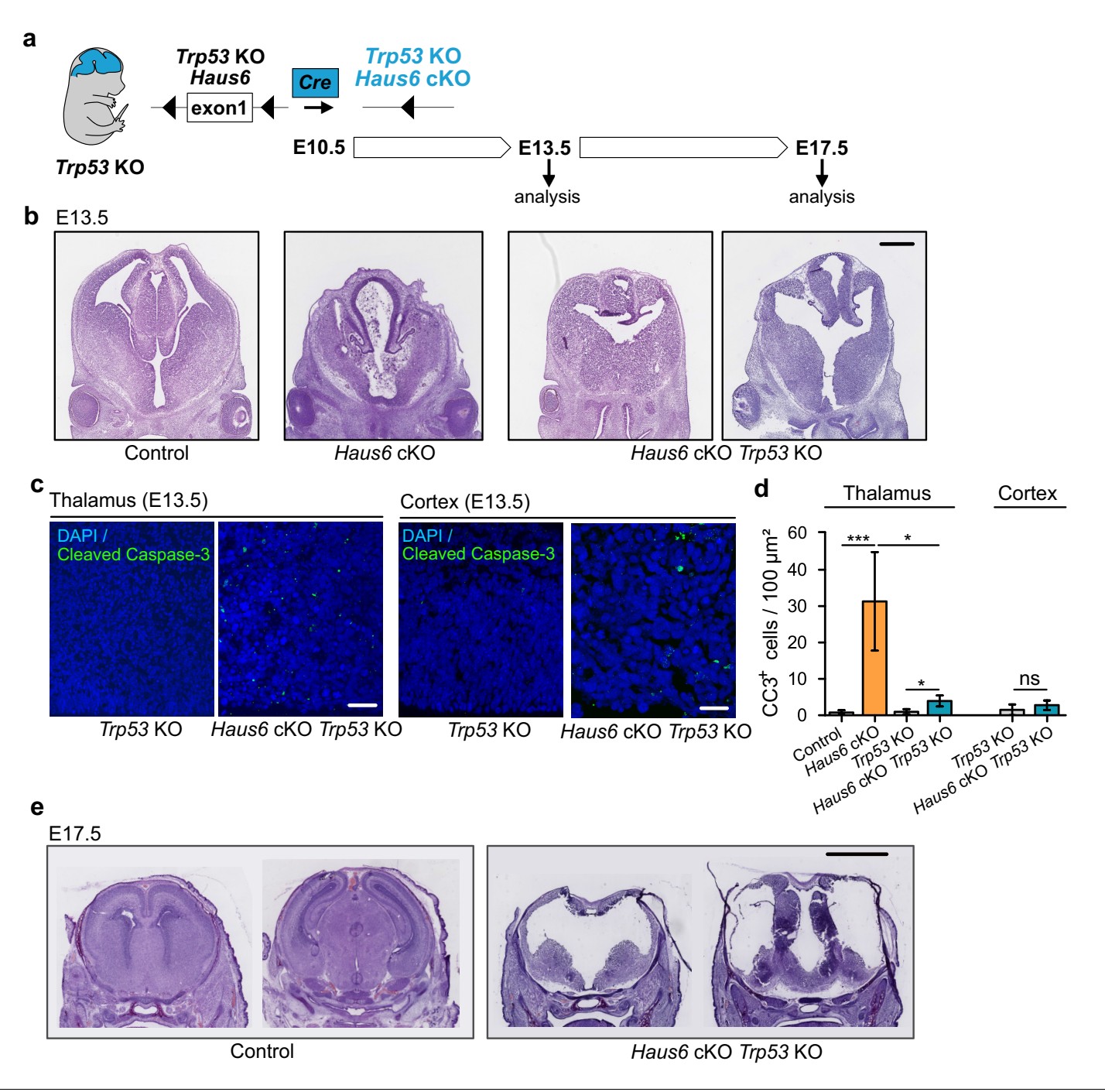

**Figure 3.** Co-deletion of *Trp53* rescues apoptosis but not abortion of forebrain development. (**a**) Schematic overview showing the experimental strategy used to generate *Haus6* cKO *Trp53* KO embryos, to test p53 dependency of the brain development phenotypes observed in *Haus6* cKO embryos. (**b**) Coronal sections of E13.5 control (*Haus6*$^{fl/wt}$ Nestin-Cre$^+$), *Haus6* cKO (*Haus6*$^{fl/fl}$ Nestin-Cre$^+$), and *Haus6* cKO *Trp53* KO (*Haus6*$^{fl/fl}$ Nestin-Cre$^+$ *Trp53*$^{-/-}$) embryos stained with hematoxylin-eosin. (**c**) Coronal sections of the thalamus and cortex of E13.5 *Trp53* KO control and *Haus6* cKO embryos stained against the apoptotic marker cleaved caspase-3 (green). DNA was labeled by DAPI (blue). (**d**) Quantification of the density of cleaved caspase-3 positive cells in the E13.5 thalamus and cortex in brain sections as shown in (**c**). n=5 for control, n=5 for *Haus6* cKO, n=4 for *Trp53* KO control, and n=3 for *Haus6* cKO *Trp53* KO embryos for quantifications in the thalamus, and n=4 for *Trp53* KO control and n=4 for Haus6 cKO *Trp53* KO embryos for quantifications in the cortex. Plotted values are means, error bars show SD. *p<0.05, **p<0.01, ***p<0.001 by two-tailed t-test. (**e**) Coronal sections of E17.5 control and *Haus6* cKO *Trp53* KO embryos stained with hematoxylin-eosin. Scale bars: (**b**) 0.5 mm; (**c**) 40 μm, 25 μm (cortex); and (**e**) 2 mm.

*Figure 3 continued on next page*

*Figure 3 continued*

The online version of this article includes the following source data and figure supplement(s) for figure 3:

**Source data 1.** Source data associated with *Figure 3d*.
**Figure supplement 1.** Expression of p21 in augmin-deficient progenitors is rescued by co-deletion of *Trp53*.
**Figure supplement 1—source data 1.** Source data associated with *Figure 3—figure supplement 1b, d*.

## Loss of p53 exacerbates mitotic defects caused by augmin deficiency

Next, we examined how co-deletion of *Haus6* and *Trp53* affected mitosis in proliferating progenitors. Similar to *Haus6* cKO alone (*Figure 2*), *Haus6* cKO *Trp53* KO embryos also had an increased density of mitotic cells in the cortex and in the thalamus as revealed by Ser10-phospho-Histone H3 staining (*Figure 4a–c*). The majority of these cells were in prometaphase (*Figure 4d*) and had disorganized spindles with fragmented spindle poles (*Figure 4—figure supplement 1a–d*). While these defects were overall similar to those observed in *Haus6* cKO brains, we also observed some differences. Centrin staining showed that ~30% of mitotic *Haus6* cKO *Trp53* KO cells had an increased number of centrioles, indicating the presence of extra centrosomes (*Figure 4—figure supplement 1e,f*). Mitotic cells with extra centrosomes had a ~2-fold increased size compared to cells with normal centrosome number (*Figure 4—figure supplement 1g*), suggesting that these cells had previously failed cytokinesis, as observed in augmin-depleted cultured cells (*Uehara et al., 2009*). Consistent with abnormal cell divisions, we also observed various abnormalities in post-metaphase cells. Compared to *Trp53* KO control littermates there was a strong increase in the number of defective anaphases and telophases including multipolar spindle configurations, lagging chromosomes, and micronuclei formation (*Figure 4e–g*). We also noticed that a fraction of *Haus6* cKO progenitors displayed enlarged nuclei in interphase (*Figure 4h*), suggesting aneuploidy/polyploidy triggered by abnormal chromosome segregation and/or failed cytokinesis. Considering that there were very few apoptotic cells in the double KO brains (*Figure 3c*), we speculated that continued proliferation may exacerbate mitotic defects. We analyzed multipolar metaphases and abnormal anaphase and telophases in *Haus6* cKO and *Haus6* cKO *Trp53* KO embryos at E13.5 in the thalamus, a structure that was present in embryos of both genotypes at this stage. We found that mitotic defects were more severe in the *Haus6* cKO *Trp53* KO brains when compared to *Haus6* cKO brains (*Figure 4i,j*).

## Mitotic errors in augmin-deficient progenitors correlate with DNA damage

Since mitotic errors can cause DNA breaks (*Quignon et al., 2007*), we probed brain tissue of *Haus6* cKO *Trp53* KO embryos for the presence of γH2AX foci, a marker of an active DNA damage response. Indeed, at E13.5 the percentage of cells with interphase nuclei displaying DNA damage was strongly increased in both the cortex and thalamus when compared to controls (*Figure 5a–d*). Side-by-side comparison of γH2AX staining in E13.5 thalamus of *Haus6* cKO and *Haus6* cKO *Trp53* KO embryos showed that augmin deficiency led to increased DNA damage relative to controls and that absence of p53 further increased this effect (*Figure 5c,d*). Thus, the extent of mitotic defects that we observed in *Haus6* cKO and *Haus6* cKO *Trp53* KO embryos was correlated with a concomitant increase in DNA damage.

## Augmin deficiency reduces neurogenesis

Centrosome defects result in premature differentiation in human cerebral organoid models (*Gabriel et al., 2016*; *Lancaster et al., 2013*). We wondered whether premature differentiation may contribute to the defects observed in augmin-deficient mouse brains. To address this, we labeled embryonic apical progenitors in S-phase by BrdU injection into pregnant mice at E12.5 and sacrificed the embryos for analysis 24 hr later (*Figure 6a*). We then determined among the BrdU-positive cells the proportion that had exited the cell cycle (negative for Ki67 staining) or that underwent neuronal differentiation (negative for PAX6 staining, positive for βIII-tubulin staining) in cortex and thalamus (*Figure 6b–e*). We observed that compared to controls the proportion of BrdU-positive, βIII-tubulin expressing cells was reduced in both *Haus6* cKO and *Haus6* cKO *Trp53* KO brains. This result suggested that mitotic defects caused by augmin deficiency did not result in premature differentiation but rather interfered with neurogenesis, and that this was not rescued by co-deletion of *Trp53*.

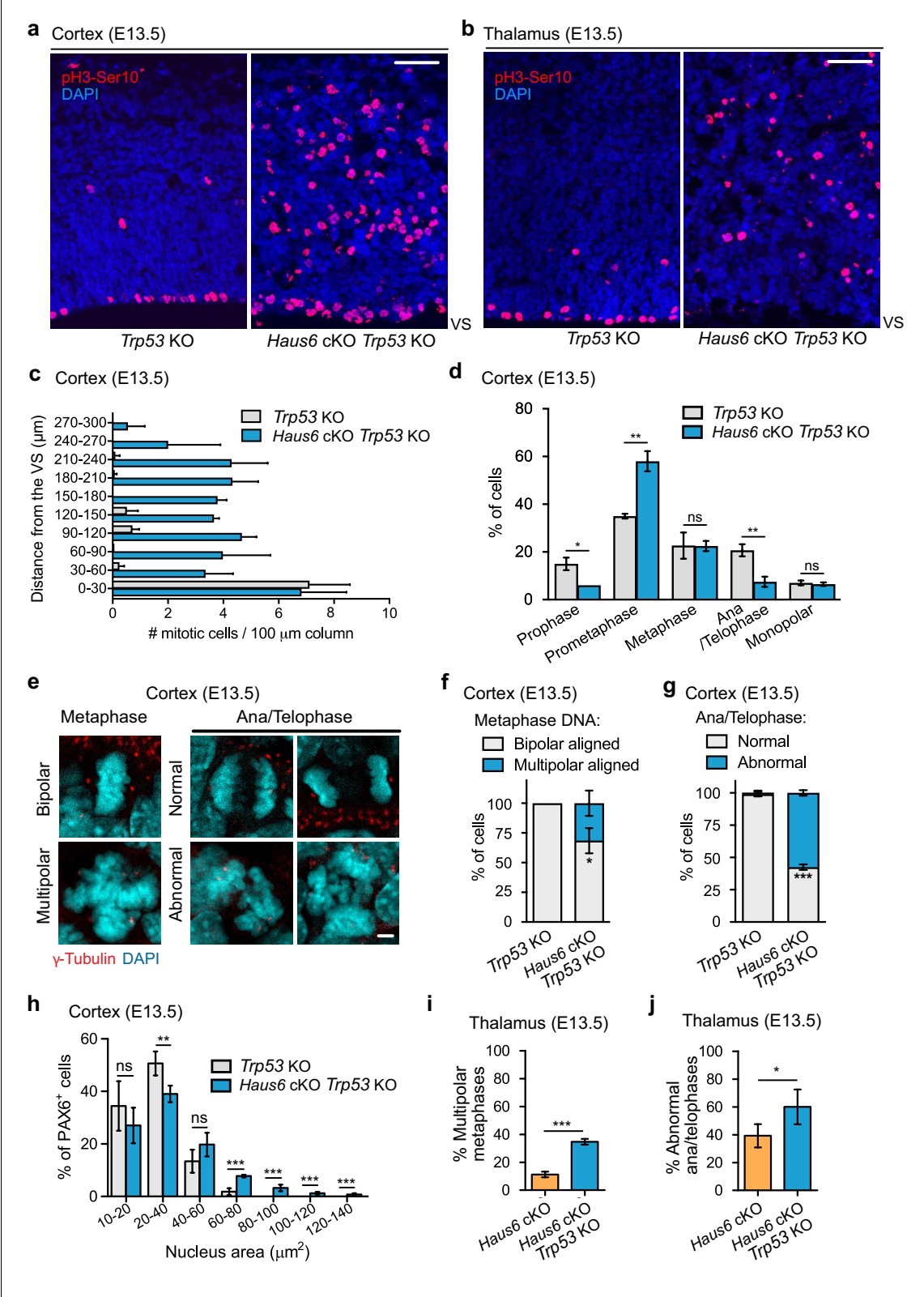

**Figure 4.** Co-deletion of *Trp53* exacerbates mitotic defects caused by augmin deficiency. (a, b) Representative coronal sections of the (a) cortex and the (b) thalamus of E13.5 *Trp53* KO control (*Haus6*[fl/wt] Nestin-Cre[-]*Trp53*[−/−]) and *Haus6* cKO *Trp53* KO embryos (*Haus6*[fl/fl] Nestin-Cre[+]*Trp53*[−/−]). Sections were stained with Ser10-phospho-Histone H3 antibody (pH3-Ser10) (red – mitotic cells) and DAPI to stain DNA (blue). (c) Quantification of the density of progenitors undergoing mitosis in the cortex at the indicated distances in μm from the ventricular surface (VS). n=3 for *Trp53* KO control and

*Figure 4 continued on next page*

Figure 4 continued

n=2 for *Haus6* cKO *Trp53* KO embryos (total of 171 and 485 cells, respectively, from four sections per embryo). (d) Quantification of mitotic progenitors at the indicated mitotic stages. n=3 for *Trp53* KO control and n=2 for *Haus6* cKO *Trp53* KO embryos (total of 442 and 443 mitotic cells counted, respectively, 140–248 cells per embryo). (e) Examples of normal, bipolar mitotic stages, and of stages with multipolar and other abnormal configurations in the cortex of E13.5 control and *Haus6* cKO *Trp53* KO embryos, respectively. Coronal sections were stained with an antibody against γ-tubulin to label spindle poles (red) and DAPI (cyan) to label DNA. (f) Quantification of the percentage of metaphase cells displaying aligned chromosomes with bipolar (white) and multipolar (blue) configuration in the cortex of embryos with the indicated genotypes. n=3 for *Trp53* KO control and n=2 for *Haus6* cKO *Trp53* KO embryos (total of 100 and 97 metaphases counted, respectively, 27–51 metaphases per embryo). (g) Quantification of the percentage of normal and abnormal ana/telophases in the cortex of embryos with the indicated genotypes. n=3 for *Trp53* KO control and n=2 for *Haus6* cKO *Trp53* KO embryos (total of 91 and 33 ana/telophases counted, respectively, 16–34 per embryo). (h) Quantification of the nucleus area in interphase PAX6-positive progenitors in the cortex of E13.5 *Trp53* KO control and *Haus6* cKO *Trp53* KO embryos. n=5 for *Trp53* KO control and n=4 for *Haus6* cKO *Trp53* KO embryos (330–2012 nuclei per embryo). (i) Quantification of the percentage of metaphase cells displaying aligned chromosomes with multipolar configuration in the thalamus of embryos with the indicated genotypes. n=4 for *Trp53* KO control and n=3 for *Haus6* cKO *Trp53* KO embryos (total of 156 and 161 metaphases counted, respectively, 32–81 per embryo). (j) Quantification of the percentage of abnormal ana/telophases in the thalamus of embryos with the indicated genotypes. n=4 for *Trp53* KO control and n=3 for *Haus6* cKO *Trp53* KO embryos (total of 103 and 90 ana/telophases counted, respectively, 17–39 per embryo). (c, d, f, g, h, i, j) Plots show mean values, error bars indicate SD. *$p<0.05$, **$p<0.01$, ***$p<0.001$ by two-tailed t-test. Scale bars: (a, b) 50 μm and (e) 5 μm.

The online version of this article includes the following source data and figure supplement(s) for figure 4:

**Source data 1.** Source data associated with *Figure 4c, d, f, g, h, i, j*.
**Figure supplement 1.** Spindle defects persist in *Haus6* cKO *Trp53* KO cells.
**Figure supplement 1—source data 1.** Source data associated with *Figure 4—figure supplement 1b, d, f, g*.

## Loss of augmin in *Trp53* KO brains disrupts neuroepithelium integrity

Apart from the aberrant mitoses in *Haus6* cKO *Trp53* KO progenitors, the distribution of mitotic figures within the tissue was also highly abnormal. Whereas in control and *Haus6* cKO brains, the vast majority of mitotic figures with condensed chromosomes were observed in the apical region, near the VS (*Figure 2a,b*; *Figure 2—figure supplement 1b,d*), in *Haus6* cKO *Trp53* KO brains most of the mitotic figures were distributed throughout the tissue including more basal regions (*Figure 4a–c*).

The presence of large numbers of basally positioned mitotic figures in the cortex and thalamus of *Haus6* cKO *Trp53* KO embryos could indicate that apical progenitors had delaminated, that their nuclei did not migrate to the apical region prior to division, or that the cells displaying mitotic defects in basal layers were not apical progenitors. The latter possibility was tested by PAX6 staining (*Figure 7a*). Whereas in the cortex of *Trp53* KO controls PAX6-positive cells were confined to the VZ, well separated from more basally positioned neurons labeled by βIII-tubulin staining, in *Haus6* cKO *Trp53* KO cortex PAX6-positive cells localized indiscriminately in basal and apical regions of the cortex, largely overlapping with regions populated by βIII-tubulin-positive neurons (*Figure 7a,d*). Interestingly, TBR2-positive intermediate progenitors, residing in the subventricular zone in control sections, had also lost this confined localization in *Haus6* cKO *Trp53* KO cortexes (*Figure 7b,e*). During development, apical progenitors in interphase maintain a bipolar structure with their centrosomes lining the VS, a configuration that is readily visualized by γ-tubulin staining in control embryos (*Figure 7c*). In *Haus6* cKO *Trp53* KO embryos, apical centrosome localization was strongly reduced and sometimes completely lost (*Figure 7c,f*). Instead, clusters of γ-tubulin foci were observed in subventricular regions, where they were never observed in controls (*Figure 7c*). Centrin staining indicated the presence of many centrioles, confirming that these were clustered centrosomes rather than PCM fragments (*Figure 7—figure supplement 1a*). Taken together, these observations suggested that progenitors in *Haus6* cKO *Trp53* KO cortexes were not only incorrectly positioned, but had also lost their polar organization. To assess this more directly, we stained for nestin, an intermediate filament protein specifically expressed in apical progenitors. In the cortex of control embryos, nestin-stained progenitors displayed a highly polarized, apicobasal morphology and a laterally aligned arrangement within the tissue (*Figure 7—figure supplement 2a*). In contrast, polarized morphology and lateral alignment were completely disrupted in progenitors of *Haus6* cKO *Trp53* KO embryos (*Figure 7—figure supplement 2a*). Consistent with these observations, staining with α-tubulin antibodies revealed that microtubules displayed apicobasal organization in control cells, running along the length of the highly polarized cell bodies (*Figure 7—figure supplement 2b,c*). In

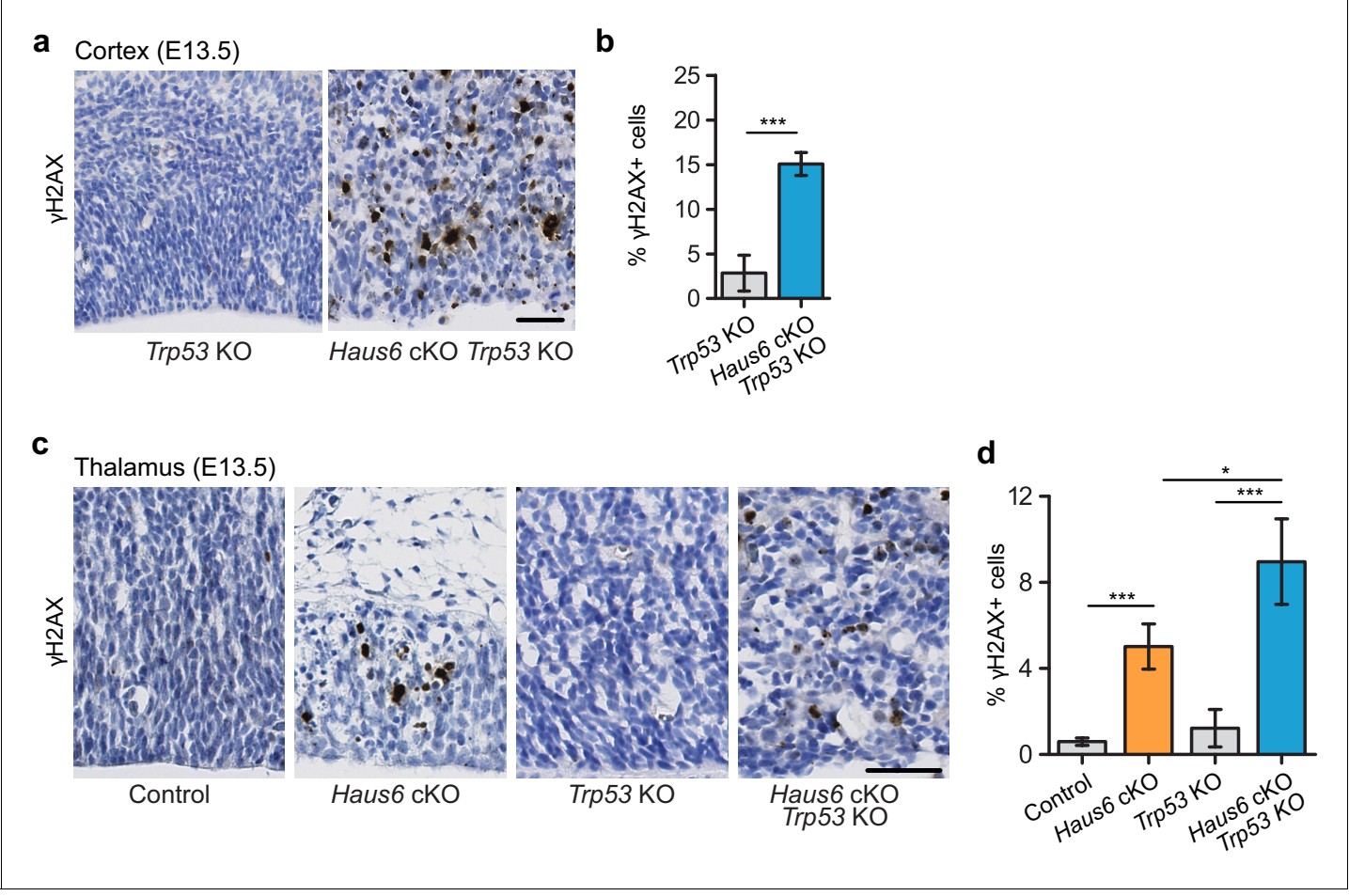

**Figure 5.** Impaired mitosis in augmin-deficient progenitors causes DNA damage. (a) Representative images of a region of the cortex of E13.5 *Trp53* KO control (*Haus6*fl/wt Nestin-Cre+ *Trp53*−/−) and *Haus6* cKO *Trp53* KO (*Haus6*fl/fl Nestin-Cre+ *Trp53*−/−) embryos. Coronal sections were stained by immunohistochemistry with an antibody against γH2AX (brown). (b) Quantification of the percentage of cells overexpressing γH2AX in the E13.5 cortex. n=4 for *Trp53* KO control and n=4 for *Haus6* cKO *Trp53* KO embryos (total of 9874 and 14,506 cells counted, respectively, two sections per embryo). (c) Representative images of the region of the thalamus of E13.5 control (*Haus6*fl/wt Nestin-Cre+ *Trp53*+/+), *Haus6* cKO (*Haus6*fl/fl Nestin-Cre+ *Trp53*+/+), *Trp53* KO control, and *Haus6* cKO *Trp53* KO embryos. (d) Quantification of γH2AX-positive cells in the E13.5 thalamus in embryos of the indicated genotypes. n=3 for control, n=4 for *Haus6* cKO, n=4 for *Trp53* KO, and n=3 for *Haus6* cKO *Trp53* KO embryos (total of 10,773, 5433, 23,384, 20,602 cells counted, respectively, two sections per embryo). (b, d) Plots show mean values, error bars indicate SD. *p<0.05, **p<0.01, ***p<0.001 by two-tailed t-test. Scale bars: (a, c) 45 μm.

The online version of this article includes the following source data for figure 5:

**Source data 1.** Source data associated with *Figure 5b, d*.

contrast, microtubules in *Haus6* cKO *Trp53* KO progenitors lacked apicobasal orientation and appeared disorganized (*Figure 7—figure supplement 2b,c*).

Taken together, these data suggest that in *Haus6* cKO *Trp53* KO embryos apical progenitors had lost their polarized organization and divided ectopically. As a result, neuroepithelium integrity was severely disrupted.

## Discussion

The mitotic spindle serves to segregate the replicated chromosomes faithfully into two daughter cells. This task is carried out by spindle microtubules and a multitude of proteins that nucleate, organize, and remodel these microtubules during mitotic progression. Here, we have analyzed the contribution of one of three different microtubule nucleation pathways, augmin-mediated microtubule amplification, to mitotic spindle assembly in proliferating neural progenitor cells during mouse brain

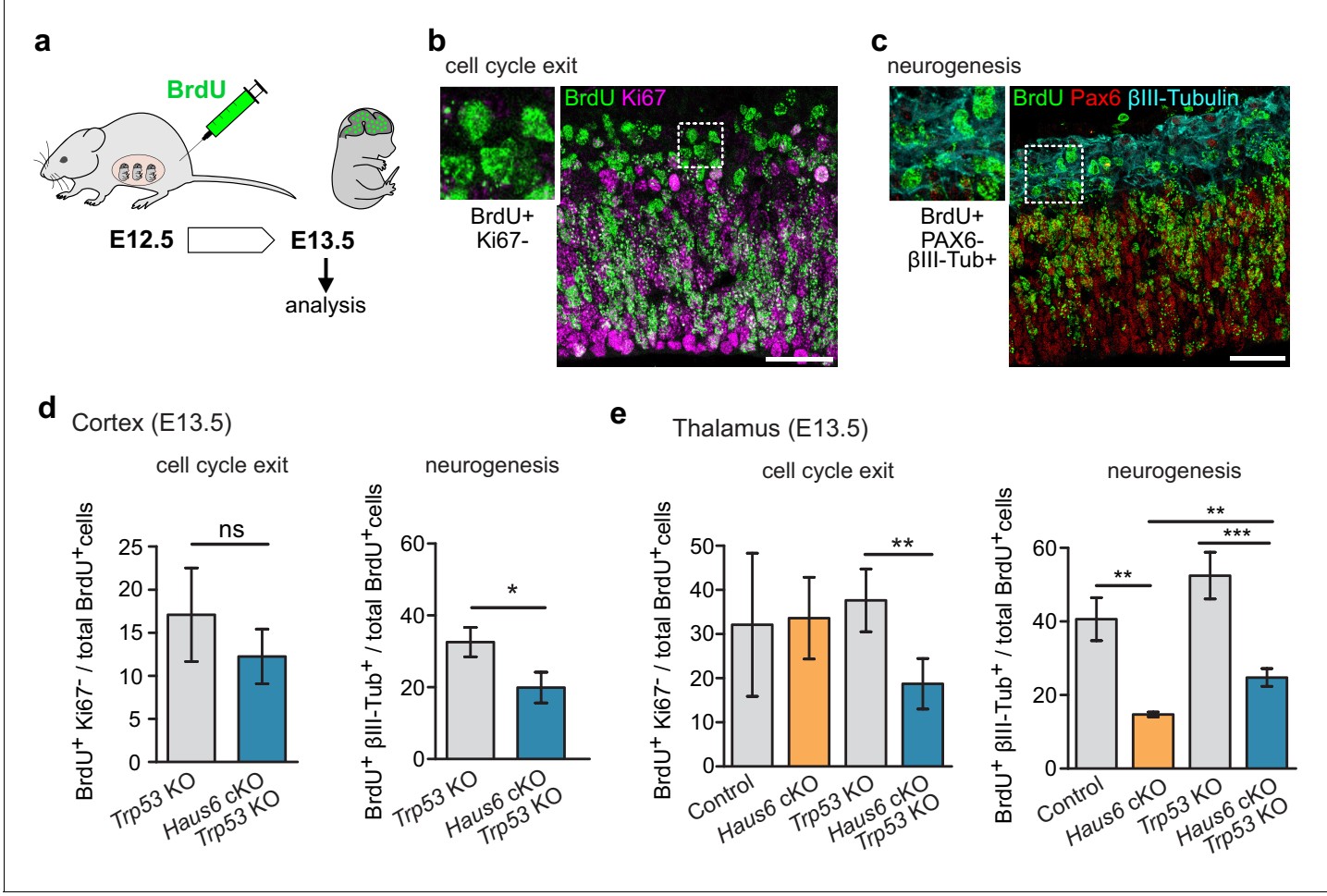

**Figure 6.** The production of neurons is reduced in *Haus6* cKO and *Haus6* cKO *Trp53* KO brains. (a) Schematic depicting the experimental procedure of BrdU injection and analysis. (b) Examples of brain sections stained with antibodies against BrdU and the proliferation marker Ki67. Identification of BrdU-labeled cells that did not display Ki67 staining is shown. (c) Examples of brain sections stained with antibodies against BrdU, the progenitor marker PAX6, and the neuronal marker βIII-tubulin. Identification of BrdU-labeled cells that did not express PAX6 but were positive for βIII-tubulin is shown. (d) Quantifications of cells in E13.5 cortex of *Trp53* KO (control) and *Haus6* cKO *Trp53* KO embryos stained as in (b and c). The percentage of BrdU-positive cells showing the indicated staining was plotted. For cell cycle exit analysis, n=4 for *Trp53* KO and n=4 for *Haus6* cKO *Trp53* KO embryos were analyzed (total of 8730 and 7361 BrdU positive cells per genotype, respectively, 1381–2942 cells in 2–3 sections per embryo). For analysis of neurogenesis, n=4 for *Trp53* KO and n=3 for *Haus6* cKO *Trp53* KO embryos were analyzed (total of 13,194 and 8139 BrdU positive cells per genotype, respectively, 1750–4614 in 3–6 sections per embryo). (e) Quantifications of cells in E13.5 thalamus of control and *Haus6* cKO, and *Trp53* KO (control) and *Haus6* cKO *Trp53* KO embryos stained as in (b and c). The percentage of BrdU-positive cells showing the indicated staining was plotted. For cell cycle exit analysis, n=3 for control, n=3 for *Haus6* cKO, n=4 for *Trp53* KO, and n=4 for *Haus6* cKO *Trp53* KO embryos were analyzed (total 9050, 5923, 12,663, 8610 of BrdU positive cells per genotype, respectively, 1159–4106 cells in 2–3 sections per embryo). For analysis of neurogenesis, n=3 for control, n=3 for *Haus6* cKO, n=4 for *Trp53* KO, and n=3 for *Haus6* cKO *Trp53* KO embryos were analyzed (total of 11,749, 5301, 8927 and 4377 BrdU positive cells per genotype, respectively, 1363–5087 cells in 2–5 sections per embryo). (d,e) Plots show mean values, error bars indicate SD. *p<0.05, **p<0.01, ***p<0.001 by two-tailed t-test. Scale bars: (b, c) 40 µm.

The online version of this article includes the following source data for figure 6:

**Source data 1.** Source data associated with *Figure 6d, e*.

development. Previous work found that impairment of centrosomal microtubule nucleation in apical progenitors slowed mitotic spindle assembly and progression, leading to p53-dependent apoptosis and causing microcephaly (*Insolera et al., 2014*; *Lin et al., 2020*; *Marjanović et al., 2015*; *McIntyre et al., 2012*; *Novorol et al., 2013*). Similarly, we found that augmin-deficiency also impaired spindle assembly, delayed mitosis, and induced p53-dependent apoptosis. In agreement with previous functional studies in cell lines (*Lawo et al., 2009*), augmin-deficient progenitors displayed fragmented spindle poles, but this did not significantly impair spindle positioning. The most

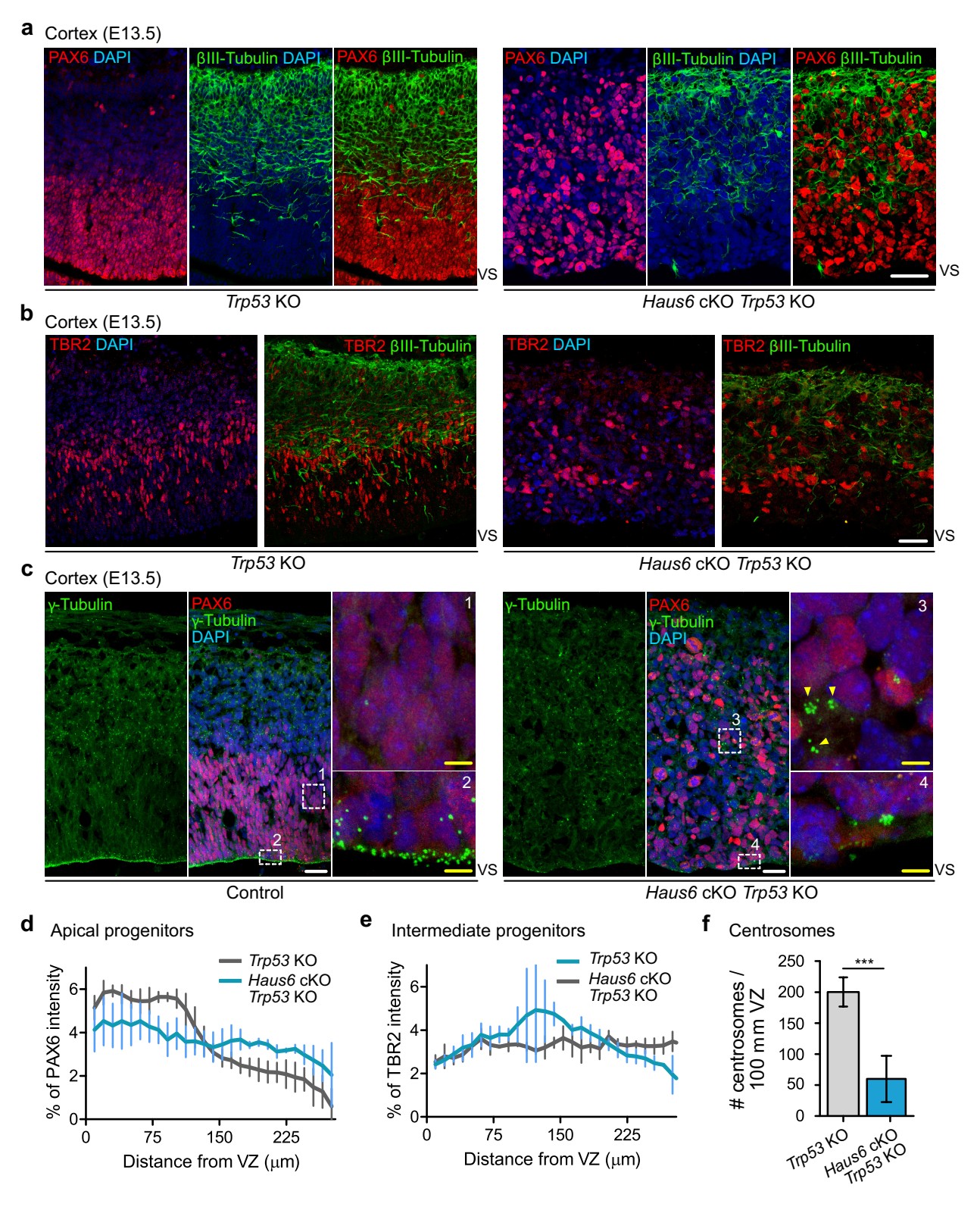

**Figure 7.** Co-deletion of *Haus6* and *Trp53* leads to loss of cortical layering. (a, b) Representative images of the E13.5 cortex from Trp53 KO control (*Haus6*^fl/wt Nestin-Cre^+*Trp53*^−/−) and *Haus6* cKO *Trp53* KO (*Haus6*^fl/fl Nestin-Cre^+*Trp53*^−/−) embryos stained with antibodies against PAX6 (a) or TBR2 (b) (red) and the neuronal marker βIII-tubulin (green). DNA was stained with DAPI. (c) Representative images of E13.5 cortex from control (*Haus6*^fl/fl Nestin-Cre^-*Trp53*^−/−) and *Haus6* cKO *Trp53* KO (*Haus6*^fl/fl Nestin-Cre^+*Trp53*^−/−) embryos. Coronal sections were co-stained with antibodies against γ-

*Figure 7 continued on next page*

# eLife Research article

Cell Biology | Developmental Biology

Figure 7 continued

tubulin (green) and PAX6 (red). DNA was stained with DAPI. Magnifications of the boxed regions labeled with 1, 2, 3, and 4 are shown. In the magnified region labeled with 4, yellow arrowheads point to ectopic clusters of interphase centrosomes. (d, e) Distribution of PAX6 and TBR2 staining in sections as in (a and b), respectively. Intensity values were averaged into 9.8-µm-thick bins and plotted as the percentage of total intensity. Lines connect mean values and error bars display SD. (d) n=5 for *Trp53* KO and n=4 for *Haus6* cKO *Trp53* KO embryos. (e) n=2 for *Trp53* KO and n=2 for *Haus6* cKO *Trp53* KO embryos. (f) Quantification of the density of centrosome number at the ventricular surface of the cortex of E13.5 *Trp53* KO and *Haus6* cKO *Trp53* KO embryos. n=4 for *Trp53* KO and n=4 for *Haus6* cKO *Trp53* KO embryos. Plots show mean values and error bars display SD. ***p<0.001 by two-tailed t-test. Scale bars: (a, b) 35 µm and (c) white – 25 µm, yellow – 5 µm.

The online version of this article includes the following source data and figure supplement(s) for figure 7:

**Source data 1.** Source data associated with *Figure 7d, e, f*.
**Figure supplement 1.** Identification of interphase centrosome clusters in *Haus6* cKO *Trp53* KO cortex.
**Figure supplement 2.** Co-deletion of *Haus6* and *Trp53* disrupts polarity in surviving progenitors.

important outcome of these defects was cell death. Our finding that the large majority of cells positive for expression of p53 and the apoptotic marker cleaved caspase-3 were PAX6-positive interphase cells, suggests that cell death occurred after completion of abnormal mitoses. Despite the similarities with centrosome defects, the *Haus6* conditional knockout phenotype is much more severe. Rather than leading to microcephaly, augmin deficiency completely aborted brain development. To our knowledge, this has not been reported for any other microtubule regulator affecting mitotic spindle assembly and progression. How can this be explained? While mitotic defects and apoptosis were also observed after loss of centrioles by conditional *CenpJ/Cpap/Sas4* knockout (*Insolera et al., 2014*) and amplification of centrosome number by PLK4 overexpression (*Marthiens et al., 2013*), the specific spindle defects caused by augmin deficiency may be a more potent trigger of apoptotic cell death than defects resulting from centrosome abnormalities. It should be noted that a more recent *Cenpj* conditional knockout mouse model displayed more severe disruption of forebrain structures, causing lethality a few weeks after birth (*Lin et al., 2020*). Still, these defects seem less severe than what we observed after augmin knockout. One may expect that preventing cell death in augmin-deficient progenitors would, at least to some degree, rescue brain development. Co-deletion of *Trp53* in *Haus6* cKO mice largely rescued apoptosis, revealing that cell death was p53-dependent, but did not rescue brain development and lethality. In the absence of apoptosis, augmin-deficient progenitors likely underwent repeated cycles of abnormal mitoses, leading to increasingly severe mitotic abnormalities. This behavior has recently been described after the induced knockout of the augmin subunit *HAUS8* in the RPE1 cell line. Whereas *HAUS8* knockout in a *TRP53* wild-type background only mildly impaired mitosis before cells arrested in G1, co-deletion of *TRP53* eliminated cell cycle arrest and exacerbated mitotic defects (*McKinley and Cheeseman, 2017*). Consistent with this possibility, *Haus6* cKO *Trp53* KO progenitors had more severe mitotic defects than *Haus6* cKO cells, including lagging chromosomes and multipolar spindles at post-metaphase stages, and displayed increased DNA damage. We have not formally tested whether cell death in augmin-deficient progenitors involves the recently described, USP28-53BP1-p53-p21-dependent mitotic surveillance pathway (*Lambrus and Holland, 2017*). However, our results show that during brain development cells that have undergone erroneous mitosis are efficiently eliminated in a p53-dependent manner, and that this occurs independently of whether the cause is centrosomal or non-centrosomal. The situation may be different in human brain development, where premature differentiation rather than apoptosis was shown to be the main response to centrosome defects in microcephaly organoid models (*Gabriel et al., 2016*; *Lancaster et al., 2013*). How human brain development would be affected by augmin deficiency is unclear. However, considering the severity of the *Haus6* KO phenotype in mice, augmin deficiency may also be lethal in humans.

The pole-fragmentation phenotype in augmin-deficient mitotic progenitors may be comparable to mitoses in the presence of extra centrosomes, as described in mice overexpressing PLK4 (*Marthiens et al., 2013*). In these animals co-deletion of *Trp53* also exacerbated mitotic defects and aneuploidy, but the outcome was still a microcephalic brain (*Marthiens et al., 2013*). In contrast, in the case of *Haus6* cKO *Trp53* KO progenitors in our study, continued proliferation was not productive for brain development. While some cortical structures were present at E13.5, they lacked a pseudostratified epithelial organization. Progenitors had lost their characteristic, highly polarized

morphology and formed a disorganized cell mass that was intermingled with βIII-tubulin-positive differentiated neurons, in both apical and basal regions. Considering that the polarized apical progenitor morphology is integral to the organization of the neuroepithelium, providing scaffold function and guidance for translocating basal progenitors and migrating neurons, it is not surprising that these defects lead to abortion of brain development.

Exacerbated mitotic errors and DNA damage as a result of continued proliferation are a reasonable explanation for the severely disrupted tissue integrity in *Haus6* cKO *Trp53* KO brains. However, we cannot exclude that additional roles of augmin contribute to this phenotype. For example, augmin may promote progenitor polarity by generating and/or maintaining the apicobasal interphase microtubule array. Recent work has shown that experimentally altered spindle positioning in progenitors can lead to loss of apical membrane. This can be compensated for by re-extension of the apical process and re-integration of the apical foot at the VS (*Fujita et al., 2020*). Assuming a role of augmin in progenitor polarity, this process may be impaired in augmin-deficient cells. Consistent with this possibility, microtubules in *Haus6* cKO *Trp53* KO progenitors appeared disorganized, lacking the apicobasal alignment that is observed in control cells. However, it is unclear whether this is cause or consequence of the loss of polarized cell morphology. It should also be noted that augmin nucleates microtubules in post-mitotic neurons, affecting their morphogenesis and their migration (*Cunha-Ferreira et al., 2018*; *Sánchez-Huertas et al., 2016*), which could contribute to tissue disruption in *Haus6* cKO *Trp53* KO brains.

In summary, our work shows that, in contrast to centrosomal nucleation, augmin-mediated microtubule amplification in neural apical progenitors is essential for brain development and cannot be compensated for by the chromatin- and centrosome-dependent nucleation pathways. As in the case of progenitors lacking centrosomal nucleation, mitotic delay caused by augmin deficiency triggers p53-dependent apoptosis. While cell death can be prevented by co-deletion of *Trp53*, the specific defects that result from the loss of augmin are sufficient to completely abort brain development, independent of p53 status.

# Materials and methods

### Key resources table

| Reagent type (species) or resource | Designation | Source or reference | Identifiers | Additional information |
|---|---|---|---|---|
| Gene (*Mus musculus*) | *Haus6* | NCBI gene | Gene ID: 230376 | |
| Strain, strain background (*M. musculus*) | Nestin-Cre *Haus6* cKO | This paper | | See Materials and methods |
| Strain, strain background (*M. musculus*) | Nestin-Cre *Haus6* cKO *Trp53* KO | This paper | | See Materials and methods |
| Strain, strain background (*M. musculus*) | *Haus6* floxed Neo (*Haus6*<sup>fl-Neo</sup>) | RIKEN http://www2.clst. riken.jp/arg/mutant %20mice%20list.html | CDB1218K | |
| Strain, strain background (*M. musculus*) | *Haus6* floxed (*Haus6*<sup>fl</sup>) | RIKEN http://www2.clst. riken.jp/arg/mutant %20mice%20list.html | CDB1354K RRID:IMSR_RBRC09630 | |
| Strain, strain background (*M. musculus*) | C57BL/6-Tg(CAG-flpe)36Ito/ItoRbrc | RIKEN (*Kanki et al., 2006*) | RRID:IMSR_RBRC01834 | |
| Strain, strain background (*M. musculus*) | B6.Cg-Tg(Nes-cre)1Kln/J | Gift from Maria Pia Cosma (originally from Jackson Laboratories) | RRID:IMSR_JAX:003771 | |
| Strain, strain background (*M. musculus*) | *Trp53*-deficient mice (B6.129S2-Trp53tm1Tyj/J) | Jackson Laboratories | RRID:IMSR_JAX:002101 | |

*Continued on next page*

*Continued*

| Reagent type (species) or resource | Designation | Source or reference | Identifiers | Additional information |
|---|---|---|---|---|
| Antibody | Anti-α-tubulin (mouse monoclonal) | Sigma-Aldrich | #3873T | IF (1:500) |
| Antibody | Anti-acetylated α-tubulin (mouse monoclonal) | Sigma-Aldrich | #T6793 RRID:AB_477585 | IF (1:500) |
| Antibody | Anti-βIII-tubulin (rabbit polyclonal) | Abcam | #ab18207 RRID:AB_444319 | IF (1:1000) |
| Antibody | Anti-βIII-tubulin (mouse monoclonal) | BioLegend | #801201 RRID:AB_2313773 | IF (1:1000) |
| Antibody | Anti-cleaved caspase-3 (rabbit monoclonal) | Novus Biologicals | #MAB835 RRID:AB_2243951 | IF (1:500) |
| Antibody | Anti-BrdU (mouse monoclonal) | Abcam | #ab8955 RRID:AB_306886 | IF (1:750) |
| Antibody | Anti-γ-tubulin (mouse monoclonal, clone TU-30) | ExBio | #ab27074 RRID:AB_2211240 | IF (1:500) |
| Antibody | Anti-γ-tubulin (rabbit monoclonal) | Sigma-Aldrich | #T5192 RRID:AB_261690 | IF (1:500) |
| Antibody | Anti-Ki67 (rabbit polyclonal) | Abcam | #ab15580 RRID:AB_443209 | IF (1:750) |
| Antibody | Anti-nestin (mouse monoclonal) | Cell signaling | #4760 RRID:AB_2235913 | IF (1:300) |
| Antibody | Anti-p53 (mouse monoclonal) | Cell signaling | #CST2524S RRID:AB_331743 | IF (1:500) |
| Antibody | Anti-PAX6 (mouse monoclonal) | BioLegend | #901301 RRID:AB_2565003 | IF (1:300) |
| Antibody | Anti-phosphorylated-Histone H3 (rabbit polyclonal) | Millipore | #06-570 RRID:AB_310177 | IF (1:1000) |
| Antibody | Anti-TBR2 (rabbit polyclonal) | Abcam | #ab23345 RRID:AB_778267 | IF (1:200) |
| Antibody | Anti-phospho-histone H2AX (Ser139) (mouse monoclonal, clone JBW301) | Millipore | #05-636 RRID:AB_309864 | IHC (1:500) |
| Antibody | Anti-p21 (rat monoclonal, HUGO291) | Abcam | #ab107099 RRID:AB_10891759 | IHC (1:500) |
| Antibody | Anti-centrin-Alexa 488 (rabbit polyclonal) | Homemade (Andrew Holland) (*Phan et al., 2021*) | | IF (1:500) |
| Antibody | Anti-mouse IgG Alexa 488 (goat polyclonal) | Life Technologies | #A11029 RRID:AB_138404 | IF (1:500) |
| Antibody | Anti-mouse IgG1 Alexa 488 (goat polyclonal) | Life Technologies | #A21121 RRID:AB_2535764 | IF (1:500) |
| Antibody | Anti-mouse IgG1 Alexa 568 (goat polyclonal) | Life Technologies | #A21124 RRID:AB_2535766 | IF (1:500) |
| Antibody | Anti-mouse IgG1 Alexa 633 (goat polyclonal) | Life Technologies | #A21052 RRID:AB_2535719 | IF (1:500) |
| Antibody | Anti-mouse IgG2a Alexa 488 (goat polyclonal) | Life Technologies | #A21131 RRID:AB_2535771 | IF (1:500) |
| Antibody | Anti-rabbit IgG Alexa 488 (goat polyclonal) | Life Technologies | #A11034 RRID:AB_2576217 | IF (1:500) |
| Antibody | Anti-rabbit IgG Alexa 568 (goat polyclonal) | Life Technologies | #A11036 RRID:AB_10563566 | IF (1:500) |
| Antibody | Anti-rabbit IgG Alexa 633 (goat polyclonal) | Life Technologies | #A21071 RRID:AB_141419 | IF (1:500) |
| Antibody | Anti-mouse IgG HRP conjugated (goat polyclonal) | Dako-Agilent | #P0447 RRID:AB_2617137 | IHC (1:500) |

*Continued on next page*

*Continued*

| Reagent type (species) or resource | Designation | Source or reference | Identifiers | Additional information |
|---|---|---|---|---|
| Antibody | Rabbit IgG polyclonal isotype control (rabbit polyclonal) | Abcam | #ab27478 RRID:AB_2616600 | IHC (1:500) |
| Antibody | Mouse IgG1 (NCG01) isotype control (mouse monoclonal) | Abcam | #ab81032 RRID:AB_2750592 | IHC (1:500) |
| Antibody | Mouse IgG2a isotype control (eBM2a) (mouse monoclonal) | Invitrogen | #14-4724-82 RRID:AB_470114 | IHC (1:500) |
| Sequence-based reagent | mAug6KO_FW | This paper | Genomic PCR primer *Haus6* | 5′-CAACCCGAGCA ACAGAAACC-3′ |
| Sequence-based reagent | mAug6KO_Rev | This paper | Genomic PCR primer *Haus6* | 5′-CCTCCCACCAA CTACAGACC-3′ |
| Sequence-based reagent | olMR1084 | This paper | Genomic PCR primer *Cre* | 5′-GCGGTCTGGCAG TAAAAACTATC-3′ |
| Sequence-based reagent | olMR1085 | This paper | Genomic PCR primer *Cre* | 5′-GTGAAACAGCAT TGCTGTCACTT-3′ |
| Sequence-based reagent | olMR7338 | This paper | Genomic PCR primer control | 5′-CTAGGCCACAGA ATTGAAAGATCT-3′ |
| Sequence-based reagent | olMR7339 | This paper | Genomic PCR primer control | 5′-GTAGGTGGAAATTCT AGCATCATCC-3′ |
| Software and algorithm | GraphPad Prism | GraphPad Software Inc | RRID:SCR_002798 | |
| Software and algorithm | QuPath | Queens University (Belfast,UK) | RRID:SCR_018257 | |
| Software and algorithm | FIJI (ImageJ) | NIH | RRID:SCR_002285 | |
| Other | Hematoxylin | Dako-Agilent | S202084 | |
| Other | 5-Bromo-2′-deoxyuridine (BrdU) | Sigma-Aldrich | B5002 | Injected peritoneally to pregnant females at a final concentration of 120 mg/kg of animal weight |
| Other | EnVision Flex Antibody Diluent | Dako-Agilent | K800621 | |
| Other | Envision Flex Wash buffer | Dako-Agilent | K800721 | |
| Other | 3-3′-diamino-benzidine | Dako-Agilent | K3468 | |
| Commercial assay or kit | Mouse on mouse (M.O.M) Immuno-detection Kit | Vector Laboratories | BMK-2202 RRID:AB_2336833 | |

## Generation and husbandry of mice

Nestin-Cre *Haus6* cKO were obtained by crossing *Haus6* floxed (*Haus6*fl) mice with B6.Cg-Tg(Nes-cre)1Kln/J mice. *Haus6* floxed Neo mice (*Haus6*fl-Neo) (Accession no. CDB1218K, http://www2.clst. riken.jp/arg/mutant%20mice%20list.html) were generated as described (*Watanabe et al., 2016*). To generate *Haus6* floxed mice (*Haus6*fl) (RBRC09630, Accession no. CDB1354K, http://www2.clst.riken. jp/arg/mutant%20mice%20list.html, *Haus6*fl-Neo mice were crossed with C57BL/6-Tg(CAG-flpe)36Ito/ ItoRbrc (RBRC01834) (*Kanki et al., 2006*). The resultant mice without the PGK-neo cassette (*Haus6* flox mice) were maintained by heterozygous crossing (C57BL/6N background). B6-Tg(CAG-FLPe)36 was provided by the RIKEN BRC through the National Bio-Resource Project of the MEXT, Japan. B6. Cg-Tg(Nes-cre)1Kln/J mice were a gift from Maria Pia Cosma (CRG, Barcelona, Spain) and previously purchased from Jackson Laboratories. To obtain Nestin-Cre *Haus6* cKO *Trp53* KO mice, mice carrying the floxed *Haus6* (*Haus6*fl) and Nestin-Cre alleles were crossed with mice lacking *p53*. *p53*-deficient mice (B6.129S2-Trp53tm1Tyj/J) were purchased from Jackson Laboratories. All the mouse strains were maintained on a mixed 129/SvEv-C57BL/6 background in strict accordance with the European Community (2010/63/UE) guidelines in the specific-pathogen-free animal facilities of the

Barcelona Science Park (PCB). All protocols were approved by the Animal Care and Use Committee of the PCB/University of Barcelona (IACUC; CEEA-PCB) and by the Departament de Territori I Sostenibilitat of the Generalitat de Catalunya in accordance with applicable legislation (Real Decreto 53/2013). All efforts were made to minimize use and suffering.

## Mice genotyping

Genotyping was performed by polymerase chain reaction (PCR) using genomic DNA extracted from tail or ear biopsies. Biopsies were digested with Proteinase-K (0.4 mg/ml in 10 mM Tris-HCl, 20 mM NaCl, 0.2% SDS, and 0.5 mM EDTA) overnight at 56°C. DNA was recovered by isopropanol precipitation, washed in 70% ethanol, dried, and resuspended in $H_2O$. To detect *Haus6* wt (800 bp), *Haus6* floxed (1080 bp), and *Haus6* KO (530 bp) alleles by PCR the following pair of primers were used: mAug6KO_FW (5′-CAACCCGAGCAACAGAAACC-3′) and mAug6KO_Rev (5′-CCTCCCACCAAC TACAGACC-3′). These PCRs were run for 35 cycles with an annealing temperature of 64.5°C. To detect the transgenic Cre-recombinase allele in Nestin-Cre cKO mice (100 bp) primers olMR1084 (5′-GCGGTCTGGCAGTAAAAACTATC-3′) and olMR1085 (5′-GTGAAACAGCATTGCTGTCACTT-3′) were used. For this PCR, primers olMR7338 (5′-CTAGGCCACAGAATTGAAAGATCT-3′) and olMR7339 (5′-GTAGGTGGAAATTCTAGCATCATCC-3′) were used as internal control (324 bp). These PCRs were run for 35 cycles with an annealing temperature of 51.7°C.

## BrdU injections

Pregnant females with embryos at E12.5 were injected intraperitoneally with 5-Bromo-2′-deoxyuridine (BrdU) (B5002; Sigma-Aldrich) diluted in phosphate-buffered saline (PBS) at a final concentration of 120 mg per kg of animal weight. After 24 hr, embryonic brain tissue was processed for histopathology analysis as described in the next section.

## Histology, immunofluorescence, and immunohistochemistry

For histopathology analysis of mouse embryos, timed pregnant female mice were euthanized and embryos were removed. Following euthanasia, embryo heads were fixed in 4% PFA diluted in PBS overnight at 4°C, followed by cryoprotection in increasing concentration of sucrose in PBS (first 15%, then 30%, with a 24 hr incubation at 4°C for each sucrose concentration), followed by overnight incubation in a 1:1 solution of 30% sucrose and OCT (Tissue-Tek). Tissues were then embedded in OCT and frozen in liquid nitrogen-cooled isopentane. For tissue histological analysis, 10-µm-thick cryosections were prepared, placed on glass slides, and processed for either hematoxylin/eosin staining using standard protocols or for immunofluorescence staining. For immunofluorescence staining, cryosections were thawed at room temperature, washed with PBS, and subjected to heat-mediated antigen retrieval in citrate buffer (10 mM citric acid) at pH 6, as required. Tissue sections were permeabilized with PBS containing 0.05% TX100 (PBS-T 0.05%) for 15 min and blocked with blocking solution (10% goat serum diluted in PBS-T 0.1%). Sections were then incubated overnight at 4°C with primary antibodies diluted in blocking solution. The next day, after washing with PBS-T 0.05%, sections were incubated for 60 min with Alexa-Fluor conjugated complementary secondary antibodies and DAPI to stain DNA. Sections were again washed with PBS-T 0.05% and mounted with Prolong Gold antifading reagent (Thermo Fisher Scientific).

For immunofluorescence stainings after BrdU incorporation sections were dried and fixed with neutral buffer formalin (HT501128-4L, Sigma-Aldrich) for 10 min. Antigen retrieval was performed using citrate buffer pH 6 for 20 min at 97°C using a PT Link (Dako-Agilent). Quenching of endogenous peroxidase was performed by incubation for 10 min with Peroxidase-Blocking Solution (S2023, Dako-Agilent). Unspecific unions were blocked using 5% of goat normal serum (16210064, Life Technologies) with 2.5% BSA (10735078001, Sigma-Aldrich) for 60 min. Blocking of unspecific endogenous mouse Ig staining was also performed using Mouse on Mouse (M.O.M) Immunodetection Kit – (BMK-2202, Vector Laboratories). Primary antibodies were diluted in EnVision FLEX Antibody Diluent (K800621, Dako-Agilent) and incubated overnight at 4°C. Secondary antibodies were diluted at 1:500 and incubated for 60 min. Samples were stained with DAPI (D9542, Sigma-Aldrich) and mounted with Fluorescence mounting medium (S3023, Dako-Agilent). Specificity of staining was confirmed by staining with rabbit IgG, polyclonal Isotype control (ab27478, Abcam), mouse IgG1, Kappa

Monoclonal (NCG01) Isotype Control (ab81032, Abcam), or a mouse IgG2a kappa Isotype Control (eBM2a) (14-4724-82 IgM, Invitrogen).

Immunohistochemistry (IHC) was performed using 7 μm cuts. Prior to IHC, antigen retrieval was performed using Tris-EDTA buffer pH 9 for 20 min at 97°C using a PT Link (Dako-Agilent). Quenching of endogenous peroxidase was performed by a 10 min incubation with Peroxidase-Blocking Solution (Dako REAL S2023). Blocking was done in M.O.M. blocking reagent (MKB-2213, Vector Laboratories), 5% of goat normal serum (16210064, Thermo Fisher Scientific) mixed with 2.5% BSA diluted in Envision Flex Wash buffer (K800721, Dako-Agilent) and with Casein solution (ref: 760-219, Roche) for 60 min and 30 min, respectively. Primary and secondary antibodies were diluted with EnVision FLEX Antibody Diluent (K800621, Dako-Agilent) and incubated for 120 min. Antigen–antibody complexes were revealed with 3-3′-diaminobenzidine (K3468, Dako-Agilent). Sections were counterstained with hematoxylin (S202084, Dako-Agilent) and mounted with Mounting Medium, Toluene-Free (CS705, Dako-Agilent) using a Dako CoverStainer. Specificity of staining was confirmed by using a mouse IgG1 isotype control (ab81032, Abcam).

## Antibodies

All antibodies are listed in the key resources table.

## Image acquisition and analysis

Histology sections stained with hematoxylin/eosin (*Figure 1c,d*; *Figure 1—figure supplement 1c*; *Figure 3b,e*) or used for IHC (*Figure 5a,c*; *Figure 2—figure supplement 2b*; *Figure 3—figure supplement 1a,c*) were imaged with the digital slide scanner Nanozoomer 2.0 HT from Hamamatsu and processed with NDP.view two software from Hamamatsu. Immunofluorescence labeled histology sections (*Figure 1e*; *Figure 2a,d,h,j,l*; *Figure 2—figure supplement 1a,b,c,f,h*; *Figure 2—figure supplement 2a*; *Figure 3c*; *Figure 4a,b*; *Figure 4—figure supplement 1a,c,e*; *Figure 6*; *Figure 7*; *Figure 7—figure supplement 1*; *Figure 7—figure supplement 2*) were imaged with a Leica TCS SP5 laser scanning spectral confocal microscope. Confocal Z-stacks were acquired with 0.5 μm or 1 μm of step size depending on the experiment and using laser parameters that avoided the presence of saturated pixels. Immunofluorescence-labeled histology sections shown in *Figure 2f* and *Figure 4e* were imaged with a Zeiss 880 confocal microscope equipped with an Airyscan. In the images shown in *Figure 2f*, for the Superresolution Airyscan mode a 63× magnification, 1.4 NA oil-immersion lens with a digital zoom of 1.8× was used. The z-step between the stacks was set at 0.211 μm. In the images shown in *Figure 4e*, for the Fast Airyscan mode a 40× magnification 1.2 NA multi-immersion lens with a digital zoom of 1.8× was used. The z-step between the stacks was set at 0.5 μm. XY resolution was set at 1588×1588. Airyscan raw data were preprocessed with the automatic setting of Zen Black. Additional image processing and maximum intensity z-projections were done in ImageJ software. In each experiment, serial brain sections from multiple animals per genotype were analyzed (details in figure legends).

Radial thickness of the thalamus was measured with ImageJ as the distance between the VS and the basal surface of this brain region in E13.5 embryos. In the same regions, radial thickness of the area occupied by PAX6 and βIII-tubulin cell populations was measured.

For mitotic density, cell counts of the thalamic/cortical wall were divided into 30 μm thick bins from the apical to basal surfaces. The number of mitotic phospho-Histone H3 positive cells was counted in each bin and normalized to the column width of the region analyzed. Mitotic density in each bin was expressed as the number of mitotic cells per 100 μm of column. Centrosome integrity in mitotic cells dividing close to the apical surface of the thalamus/cortex was analyzed by quantifying the percentage of cells displaying unfocused/fragmented spindle poles, each composed of multiple γ-tubulin dots. Mitotic spindle integrity was analyzed in cells dividing close to the apical surface of the thalamus and the percentage of cells displaying abnormal, non-bipolar organized spindles were quantified.

To evaluate p53 expression, cell death, DNA damage, cell cycle exit, and neurogenesis in the embryonic forebrain, representative images of the thalamus/cortex containing the entire apicobasal axis of the tissue were selected. The number of p53 and cleaved caspase-3 positive cells was counted and divided by the area of the selected region. To evaluate the cell population overexpressing p53 in the thalamus, coronal sections were co-stained against p53, PAX6, and βIII-tubulin. Cells

in which the p53-positive nucleus was costained with PAX6 were counted as PAX6-positive. Cells with the p53-positive nucleus that did not stain for PAX6 and were surrounded by a cytoplasmic βIII-tubulin signal were considered as βIII-tubulin-positive. To evaluate the expression of phosphorylated Histone H2AX, image files obtained with the Nanozoomer 2.0 HT slide scanner were opened with the image analysis software QuPath (*Bankhead et al., 2017*). The number of phosphorylated Histone H2AX-positive cells was divided by the total amount of hematoxylin-stained cells in the specific tissue, counted using the QuPath software. To evaluate cell cycle exit and neurogenesis, embryonic tissue sections obtained from pregnant females injected with BrdU were co-stained with BrdU and Ki67 antibodies (for cell cycle exit analysis) or BrdU, PAX6, and βIII-tubulin antibodies (for neurogenesis analysis). To evaluate co-expression of the different markers, image files obtained with the Nanozoomer 2.0 HT slide scanner were opened with the image analysis software QuPath and the 'Positive Cell Detection tool was used'. Cell cycle exit was analyzed by determining the number of BrdU-positive cells that did not stain for the cell cycle marker Ki67 relative to the total number of BrdU-positive cells in the respective tissue. Neurogenesis was evaluated by determining the number of BrdU-positive cells that were also positive for βIII-tubulin staining but negative for PAX6 staining relative to the total number of BrdU-positive cells in the respective tissue. In all experiments, for each brain, at least two coronal tissue sections were quantified.

To measure interphase nucleus size in cortical neural progenitors (*Figure 4h*), tissue sections were immunostained with PAX6 antibodies and DAPI to label DNA. The area of nuclei in PAX6-positive cells in the cortex was measured in z-stack images using the 'Positive cell detection' tool of QuPath software. Mitotic cells were excluded from this analysis.

To quantify the distribution of neural progenitors within the cortex, cryosections providing lateral views of the cortex were immunostained against PAX6 or TBR2. In both cases, the 'Plot Analysis' tool of ImageJ was used to measure signal intensity along the apicobasal axis of the cortex. Measurements were grouped into 9.8-μm-wide bins and the average value for each bin was plotted as the percentage of the sum of all bin intensities.

For analysis of mitotic spindle orientation, cryosections providing coronal views of the thalamus/cortex were immunostained with DAPI and the mitotic DNA marker phosphorylated-Histone H3 and the centrosome/spindle pole marker γ-tubulin. The orientation of the mitotic spindle was then determined by measuring the angle between the pole-to-pole axis and the ventricular lining.

## Statistics

All graphs with error bars are presented as means with standard deviation. To determine statistical significance between samples, an unpaired two-way Student's t-test was used. Statistical calculations and generation of graphs were performed in Excel or Graphpad Prism6 (ns=not significant, *p<0.05, **p<0.01, ***p<0.001).

## Acknowledgements

The authors are grateful to Gohta Goshima (Nagoya University, Japan) for generously providing floxed *Haus6* mice that were generated in his laboratory and for feedback on the manuscript. The authors acknowledge excellent support by the IRB Barcelona Histopathology and Advanced Digital Microscopy core facilities for help with sample preparation and analysis, and by the Mouse Mutant Core facility for deriving floxed *Haus6* mice from sperm samples. The authors thank Travis Stracker (NIH-NCI, Bethesda) for mouse cage space and discussion, Eduardo Soriano and Antoni Parcerisas (University of Barcelona, Spain) for technical help and discussions, Andrew Holland (Johns Hopkins University, Baltimore) for anti-centrin antibodies, Pia Cosma (CRG, Barcelona, Spain) for providing Nestin-Cre mice, and Irina Matos (The Rockefeller University, New York) for helpful comments on the manuscript.

## Additional information

### Competing interests

Jens Lüders: Reviewing editor, *eLife*. The other authors declare that no competing interests exist.

## Funding

| Funder | Grant reference number | Author |
| --- | --- | --- |
| Ministerio de Ciencia, Innovación y Universidades | BFU2015-69275-P | Jens Lüders |
| Ministerio de Ciencia, Innovación y Universidades | PGC2018-099562-B-I00 | Jens Lüders |
| Agència de Gestió d'Ajuts Universitaris i de Recerca | 2017 SGR 1089 | Jens Lüders |
| Ministerio de Ciencia, Innovación y Universidades | RED2018-102723-T | Jens Lüders |
| Ministerio de Ciencia, Innovación y Universidades | SVP-2014-068770 | Ricardo Viais |
| Ministerio de Ciencia, Innovación y Universidades | PRE2019-089526 | Marcos Fariña-Mosquera |
| Ministerio de Ciencia, Innovación y Universidades | BES-2016-076423 | Marina Villamor-Payà |
| Japan Society for the Promotion of Science | 15H06270 | Sadanori Watanabe |

The funders had no role in study design, data collection and interpretation, or the decision to submit the work for publication.

## Author contributions

Ricardo Viais, Data curation, Formal analysis, Investigation, Visualization, Writing - review and editing; Marcos Fariña-Mosquera, Data curation, Formal analysis, Investigation, Visualization, Performed a subset of the stainings and quantifications of cryosections and assembled the corresponding graphs and figure panels; Marina Villamor-Payà, Methodology, Set up breedings and obtained embryos for a subset of animal experiments, and performed BrdU injections; Sadanori Watanabe, Conceptualization, Methodology, Designed the strategy for generating the conditional Haus6 knockout and generated floxed Haus6 mice; Lluís Palenzuela, Methodology, Assisted with setting up of animal breedings and maintenance of mouse colony; Cristina Lacasa, Formal analysis, Supervision, Supervised animal experiments and compliance with animal welfare regulations at IRB Barcelona, and assisted in mouse genotyping; Jens Lüders, Conceptualization, Formal analysis, Supervision, Funding acquisition, Writing - original draft, Project administration

## Author ORCIDs

Ricardo Viais (iD) https://orcid.org/0000-0002-8146-3693
Marcos Fariña-Mosquera (iD) https://orcid.org/0000-0002-3822-4718
Marina Villamor-Payà (iD) http://orcid.org/0000-0002-7288-4197
Lluís Palenzuela (iD) http://orcid.org/0000-0003-0295-5225
Jens Lüders (iD) https://orcid.org/0000-0002-9018-7977

## Ethics

Animal experimentation: All the mouse strains were maintained on a mixed 129/SvEv-C57BL/6 background in strict accordance with the European Community (2010/63/UE) guidelines in the Specific-Pathogen Free (SPF) animal facilities of the Barcelona Science Park (PCB). All protocols were approved by the Animal Care and Use Committee of the PCB/University of Barcelona (IACUC; CEEA-PCB) and by the Departament de Territori I Sostenibilitat of the Generalitat de Catalunya in accordance with applicable legislation (Real Decreto 53/2013). All efforts were made to minimize use and suffering.

## Decision letter and Author response

Decision letter https://doi.org/10.7554/eLife.67989.sa1
Author response https://doi.org/10.7554/eLife.67989.sa2

## Additional files

### Supplementary files
• Transparent reporting form

### Data availability
All data generated or analyzed during this study are included in the manuscript and supporting files.

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
