## [Decision Letter]

[Editors' note: this paper was reviewed by Review Commons.]

**Acceptance summary:**

This manuscript describes a role for augmin complex during brain development. Augmin complex recruits γ-tubulin ring complex (γTuRC) to microtubule lattices to nucleate microtubule branches. The authors show how loss of Haus6, a part of the augmin complex, in neural progenitors, leads to elevated p53 activity and apoptosis, with severe consequences on overall brain development. In particular, augmin-deleted neural progenitors display spindle abnormalities and mitotic delay, which induce DNA damage accountable for p53-induced apoptosis.

---

## [Author Response]

We thank all three reviewers for their very useful and constructive comments. Below is our point-by-point response.

Reviewer #1Evidence, reproducibility and clarity:The manuscript by Viais R et al. describes a novel role for augmin complex in apoptosis prevention during brain development. Augmin complex recruits g TuRC to microtubule lattices to nucleate microtubule branches. The authors show how -in its absence- neural progenitors have elevated p53 activity and apoptotic rate, with severe consequences on overall brain development. In particular, augmin-deleted neural progenitors display spindle abnormalities and mitotic delay, which induce DNA damage accountable for p53-induced apoptosis.One point that I personally found very interesting is the role of augmin-dependent MT nucleation depletion in interphase. The authors mention (line 152) that at stage E13.5, besides the number of neurons being reduced, a few neurons were misplaced in the apical region, indicating a role for augmin-driven MT nucleation in cell migration. Moreover, the authors showed that p53 genetic deletion in the Haus6 cKO rescues the apoptosis phenotype but not the tissue disorganisation, suggesting that augmin-dependent microtubule might play a role in tissue polarity. While this is well presented in the discussion, the title in line 268 narrowly refers to mitotic augmin roles. I would like here to see the authors referring to putative roles for augmin-mediated MT nucleation in interphase, by toning down the title in line 268.

We note that severe loss of tissue integrity is evident in the *p53* KO background. In this background cells are allowed to repeatedly undergo defective cell divisions with aberrant chromosome segregation, producing increasingly abnormal daughter cells that may eventually fail to support epithelial integrity. Regarding possible neuronal migration defects, this has been previously observed in a study by the Hoogenraad group (Cunha-Ferreira et al., 2018) and this is mentioned in our discussion. To account for the possibility that augmin may have roles beyond mitosis, we have changed the heading to a more neutral statement, not specifically referring to proliferation/mitosis: **“**Loss of augmin in *p53* KO brains disrupts neuroepithelium integrity”.

Overall, the text is well written and flows easily. Figures are clear and legends provide sufficient information on experimental conditions, number of replicates and scale bars. I noticed that, although the number of repeats is specified, the number of cells scored per experiment is not always included. In my comments below I highlight cases where this missing information should be added.Specific points:1. In the Cep63 KO (Marjanovic et al., 2015) and the CenpJ KO mice (Insolera et al., 2014), as well as other recently published papers (e.g. Phan TP et al., EMBO Journal, 2020) part of the phenotypical characterisation of the KO mice displays pictures of the overall brain dissected from the mice. Could the author show these images?

The main difference between the cited studies (including our own, Marjanovic et al., on the role of CEP63 in brain development) and our current study is that in the previous studies brains are microcephalic but essentially intact, whereas in our current study brain development was aborted and accompanied by cell death and severe tissue disruption. As a result, in many cases these brains are very fragile and difficult/impossible to isolate. An additional challenge is the fact that brain disruption occurs at a very early developmental stage (before E13.5), where dissection is more difficult than at later stages. We note that all the brains presented in the above cited studies were from later embryonic stages or newborn/adult mice. Therefore, instead of dissecting brains, we decided to present encephalic coronal and sagittal sections as shown in Figure 1c, d, e, Figure 1-supplement 1c, and Figure 3b, e to show the overall impact of *Haus6* cKO and *Haus6* cKO *p53* KO on embryonic brain morphology at E13.5 and E17.5.

2. Fig2d: do the insets correspond to higher magnification images? What is the zoom factor? I could not find it in the legend.

The zoom factor is 1.4 – we have added this information to the figure legend.

3. Fig2E,I and K graphs: how many cells were quantified here over how many experiments? I could not find information in the figure legend.

For all quantitative data, we have added information regarding the number of embryos and counted cells to the figure legends.

4. The impact of Haus6 on mitotic spindle needs further clarification:– Fig2F: here, the authors show quantification for abnormal and multipolar spindle together. Later on, the abnormal spindle phenotype is no longer discussed (Figure 4). I was wondering what is the individual contribution of abnormal and multipolar spindle, separately. Which one of the two is more frequent? Could the authors explain in the text how they define an abnormal spindle? Is it the lack of MT with the condensed chromosome area?

We agree that our previous classification was somewhat confusing. The spindle defects in *Haus6* cKO cells are directly linked to the spindle pole fragmentation phenotype shown in Figure 2d, e. Association of spindle microtubules with these scattered PCM fragments causes spindles to appear overall disorganized. In some cases, multiple smaller asters are present, which is what we had termed “multipolar”. However, this does not always involve multipolar DNA configurations, which we separately quantify in Figure 4. To avoid confusion, we now classify spindle morphologies based on tubulin staining simply as “normal” (bipolar configuration, two robust and focused asters) or “disorganized” (lack of bipolar configuration, in some cases multiple smaller asters). We have also included a better description of this classification (lines 204-212).

– Could it be that augmin deletion induce an instability in MTs within the mitotic spindle, leading to the "empty" or with very few MTs spindles? Or could it be that more cold-sensitive MTs are affected by fixation? What is the percentage of the spindle with no MT in control?

Yes, it is likely that augmin-deficient spindles are less well preserved during fixation due to compromised spindle microtubule stability. Indeed, in tissue culture cells augmin deficient spindle microtubules are more depolymerization-sensitive than controls (Goshima et al., 2008; Lawo et al., 2009; Zhu et al., 2008). We have quantified this effect and found that the percentage of mitotic cells lacking spindle microtubule staining is indeed increased in *Haus6* cKO brains (Figure 2h, i).

– Did the authors quantify anaphase/telophase phenotypes as they did in Fig4f?

Yes, this quantification was already included in Figure 4j, where we compared abnormal chromosome configurations between *Haus6* cKO and *Haus6* cKO *p53* KO.

– How do authors explain PCM fragmentation here? Could this phenotype be due to an initial cytokinesis defect which led the cells to accumulate extra centrosomes? Or could this maybe be a product of aberrant PCM maturation/centrosome duplication? Could the authors add here a line to discuss the possible origin of pole fragmentation?

The PCM fragmentation phenotype has previously been described after augmin RNAi in cultured cells (Lawo et al., 2009). We refer to this result and we have added the above reference, to emphasize this point. The authors showed that this phenotype does not involve amplification of centriole number, but is caused by an imbalance in microtubule-dependent forces acting on the PCM and leading to its fragmentation. Thus, the extra poles were formed by acentriolar PCM fragments. We have clarified this issue by quantifying centriole numbers in mitotic cells (when centriole duplication is complete) in control and *Haus6* cKO brains. This confirmed the data previously obtained in cell lines and showed that the fragmented spindle poles after *Haus6* cKO are not due to extra centrioles (Figure 2-supplement 1f, g) (see also below).

Apart from the PCM fragmentation phenotype that does not involve changes in centriole number, previous work in cultured cells also described cytokinesis defects (Uehara et al., 2009). Failed cytokinesis would indeed lead to increased centriole number. However, it would also increase DNA content, which would be visible by an increase in the size of interphase nuclei. We observed this in *Haus6* cKO *p53* KO cells, which can undergo repeated divisions in the absence of HAUS6. These data were presented in our previous manuscript version (Figure 4h). We have now quantified centriole numbers and found that these were increased in a subset of mitotic cells in *Haus6* cKO *p53* KO brains. As predicted and consistent with cytokinesis failure, these cells have an increased size compared to controls with normal centriole number. The new data are presented in Figure 4-supplement 1e,f,g.

5. Figure 4 Did the authors quantify centrosome fragmentation and abnormal spindle here? As they characterised them for the Haus6 cKO mouse, it would be preferable to maintain the same characterisation for the Haus6 cKO p53KO.

We have quantified pole fragmentation and spindle defects as shown for *Haus6* cKO in Figure 2 also for *Haus6* cKO *p53* KO. The new data are presented in Figure 4-supplemement 1.

6. Figure 4C and d: how many replicates were done to obtain these graphs? I think the authors forgot to add this information in the figure legend.

This information has been included in the figure legend.

7. Fig4f,g, I and J: how many cells were counted per experiment? I appreciate the authors writing the n of experiments performed.

We have added this information to the figure legend.

8. Fig5d: how many cells were counted per experiment?

We have added this information to the figure legend.

Significance:While it was already known that mitotic delay affects the neuronal progenitor pool through activation of p53-dependent apoptosis (Pilaz L-J, Neuron 2016; Mitchell-Dick A, Dev Neurosci 2020), and that this can be triggered by depletion of centrosomal proteins as Cenpj and Cep63, the role of surface-dependent microtubule nucleation was not identified so far. Some insights come from a Haus6-KO mouse model which dies during blastocyst stage after several aberrant mitosis (Watanabe S, Cell Reports, 2016). In parallel, McKinley KL et al. showed that Haus8 depletion in human cells (RPE1cells) triggered p53-dependent G1 arrest following mitotic defects (McKinley KL, Developmental Cell, 2017). Building on the Hause6 KO mouse and human cell line data, here Viais R et al. discover a novel role for the augmin-mediated MT nucleation in neural progenitor growth and brain development in vivo, through prevention of p53-induced apoptosis.Specifically, Viais R et al. show that:1. Surface-dependent microtubule nucleation depletion severely impacts brain development, disrupting partly or completely forebrain domains and cerebellum;2. Surface-dependent microtubule nucleation depletion induce spindle abnormalities, resulting in mitotic delay in apical progenitors;3. Mitotic delay results in DNA breaks, p53 activation and p53-induced apoptosis.This is a tidy, well-executed study with good quality data. These findings propose a novel mechanism that results essential for neural progenitor and overall brain development.In my opinion, a large audience will benefit from these discoveries: from developmental biologists to cell biologists focused on microtubule dynamics, cell cycle, differentiation, stem cells and cell polarity.Key works describing my area of expertise: microtubule dynamics, centrosome function, cell cycle regulation and cell polarity.Reviewer #2Evidence, reproducibility and clarity:Viais, Lüders and colleagues here present an analysis of augmin's roles in neural stem cell development. They describe a dramatic impact of the conditional ablation of Haus6 on embryonic brain development in the mouse, with mitotic problems that lead to greatly-increased levels of apoptosis. The rescue of this apoptosis by mutation of the gene that encodes p53 did not restore brain development, which was still aberrant, due to mitotic errors.The paper is clearly written, with well-designed and controlled experiments. Its conclusions are well supported by the data presented. I have few comments on the technical aspects of the work- it appears very solid to me.Specific comments1. Clearer explanation of the mouse strains used should be provided. The section describing the generation of the Haus6 conditional on p.5 should specify that this is the same as was already published in the 2016 Watanabe paper (this is in the Materials and methods), but this should be more clearly specified. More specific details of the p53 knockout mice from Jackson should be included in the Materials and methods.

We have included additional information describing the generation of the *Haus6* cKO mice in the text (lines 135-138). It is not exactly the same as described in the Watanabe et al. paper. The previously published strain (Watanabe et al., 2016) contained a floxed *Haus6* cKO allele with a flanking neomycin cassette. For the current study the neomycin cassette was removed. Details are described in the method section and also shown in Figure S1a. Specific information regarding the *p53* KO strain has been added to the method section.

2. Figure 1a contains minimal information on the Haus6 locus. More detail should be included for information, if this Figure is to remain (although reference to the targeting details in the original description would be sufficient). It is unclear what the timeline diagram is to convey and it should be improved or deleted. A similar comment applies for the details in Figure 3a, although the colour scheme for the different genotypes is useful.

More detailed information on the *Haus6* locus is shown in the schematic of Figure 1-supplement 1a and in the referenced study (Watanabe et al., 2016). Since the targeting of *Haus6* exon1 was previously described, we believe that including this information as a supplementary figure and referring to the previous study is appropriate.

Regarding the schematics in Figure 1a and Figure 3a, we have improved these. The timeline shows the time points of Cre expression and of obtaining embryos for analysis.

3. The important PCR controls in Figure S1b have an unexplained 1000 bp band that appears only in the floxed heterozygote. It would be helpful if the authors explained this in the relevant Figure legend.

This band is an artefact and likely represents heteroduplexes of floxed (1080 bp) and wild type (530 bp) DNA strands due to extended regions of complementary. We have explained this in the figure legend.

4. Assuming the putative centrosome 'clusters' in Figure 6c are similar to the fragmented structures seen in thalamus in Figure 2d, a different description should be used to avoid confusion with multiple centrosomes, which is not a phenotype here. It is not clear how the loss of centrosomes from the ventricular surface was scored, whether it was based on total γ-tubulin signal or individual centrosomes; how fragmented poles would affect that is unclear, so the legend and relevant details should clarify this point.

The fragmented spindle poles shown in Figure 2d are different from the centrosome clusters in Figure 6c (now Figure 7c). The fragmented poles are fragments of PCM rather than extra centrosomes. Fragmentation is specific to mitosis, involving forces exerted by spindle microtubules (Lawo et al., 2009). In contrast, the centrosome clusters that we observed in *Haus6* cKO *p53* KO apical progenitors represent centrosomes from multiple cells in interphase, most likely as part of apical membrane patches that have delaminated form the ventricular surface. In the intact epithelium of controls these centrosomes line the ventricular surface. To avoid confusion, we now indicate in the text and legend that these centrosome clusters involve interphase cells. In addition, using centrin staining, we now show that these clusters contain multiple centrioles (Figure 7-supplement 1), in contrast to the acentriolar PCM fragments in mitotic cells.

5. Phospho-histone H2AX should be referred to as a marker of activation of the DNA damage response, rather than DNA repair.

We have changed the text accordingly.

Minor pointsi. Figure 1b should include a scale bar.

We have added the scale bar.

ii. The labelling of Figure 1f should be revised.

The labels have been fixed.

iii. Figure 2k is not labelled in this Figure.

This has been fixed.

iv. Scale bars should be included in the blow-ups in Figure 6c.

We have added the scale bars.

Significance:While it is striking that they see complete disruption of brain development, rather than microcephaly, arguably the mechanistic novelty of the findings is moderate, in that the impacts of Haus6 deficiency on mitotic spindle assembly are well established. The authors only allude to potential additional and novel activities of augmin (in neural progenitors, potentially) that might explain this possibly-unexpected outcome of this study.The topic is likely to be of interest to people in the field of mitosis, genome stability and brain development.My expertise is cell biology/ mitosis, less so on murine brain development.Reviewer #3Evidence, reproducibility and clarity:Jens Lüders and Co demonstrates the essential role of Augmin-mediated MT is critical for proper brain development in mice. The most striking point is that even p53 is eliminated, the microcephaly phenotypes of Haus6 KOs were not rescued. This could mean that the Augmin-mediated MT process is critical to cellular functions that are independent of p53. The authors claim that there are increased DNA damage and excessive mitotic errors. In these aspects, the current work is fascinating.Nevertheless, what causes massive damage to the neural epithelial tissues in the double mutant is not well explained or examined. Few questions appear in mind before I go into the detail. Are these animals still harbor functional centrosomes and their numerical status?

This is an important point that was also raised by the other reviewers. Based on previous work in cells lines (Lawo et al., 2009), we do not expect that loss of augmin directly impairs centrosome number or MTOC activity. Indeed, Lawo et al. showed that centriole number was unaffected. The only centrosome defect that the authors observed was fragmentation of the PCM during mitosis, but this was shown to be due to imbalanced forces exerted by spindle microtubules: fragmentation could be rescued by microtubule depolymerization or depletion of the cortical microtubule tethering factor NUMA. We have now examined this issue also in our mouse model by staining and counting of centrioles in mitotic apical progenitors of control and *Haus6* cKO embryos. This confirmed that centriole number is not increased by *Haus6* cKO (Figure 2-supplement 1f, g).

The microcephaly part of the introduction needs some more work. In particular, the authors need to explain apical progenitors' depletion, possibly the correct mechanisms in causing microcephaly. By saying cortical progenitors, it becomes vague. Indeed, there would also be cortical progenitors depleted. But, the fundamental mechanisms are the depletion of apical progenitors lined up at VZ's lumen. Two works in this connection generated brain tissues from microcephaly patients carrying mutations in CenpJ and CDK5RAP2 (Gabriel and Lancaster et al). Authors should cite their work and relate their findings to mouse brain data.

We have introduced text changes in the introduction to indicate the specific role of apical progenitor depletion in microcephaly and the differences in the underlying mechanism between mouse and human organoid models (line 61; lines 87-90). In this context we also cite the Gabriel et al. and Lancaster et al. studies.

-What makes me worry is, looking at figure 1E, there is pretty much no brain, and of course, authors have analyzed what is left over. How could one distinguish reduced PAX6 area and TUJ1 area is due to the gross defects in brain development. Clearly, Haus6 KO causes a severe defect in brain development. Thus, deriving a conclusion from the damaged brain can be misleading. One way to circumvent this problem is to perform 2D experiments with isolated cell types (let us say NPCs and testing if they can spontaneous differentiate).

We note that overall brain structures are only lost by E17.5, but brain structures (albeit defective) are still present at E13.5. Indeed, all of our quantifications were done at E13.5 or earlier stages. That being said, we understand the concern that quantifications in defective brain structures may be misleading. However, 2D cultures, for which cells are removed from their tissue context, may have similar issues. For this reason, we have performed BrdU injection experiments. 24 hours after incorporation of the label by proliferating apical progenitors during S phase, we fixed embryos and determined the proportion of BrdU-positive cells that had stopped cycling (Ki67 negative), expressed the progenitor marker PAX6, or were positive for the neuronal marker βIII-tubulin. The data show that there is no significant fraction of cells undergoing premature differentiation (new Figure 6).

Independently of the BrdU experiments, we have also stained brain sections with antibodies against the cell cycle inhibitor p21. Expression of p21 was increased in a fraction of cells in *Haus6* cKO brains (Figure 2-supplement 2b,c) and this was rescued in Haus6 cKO *p53* KO brains (Figure 3-supplement 1a-d). Together the data confirm apoptosis or cell cycle arrest, but not premature differentiation, as main responses to mitotic errors after *Haus6* cKO.

Figure 2: A nice illustration that Hau6 KO animals harbor many mitotic figures. The quantifications lack how many slices and how many cells were analyzed. Simply n=4 does not say much. 4 animals were considered but how many cells/slices would help identify mitotic cells/animals' distribution. A simple bar diagram does not tell a lot.

We have added this information for all quantifications to the figure legends.

As a minor point, how did the authors unambiguously scored prometaphase cells and other mitotic figures? Representative figures will help. Besides, what is the meaning of many prometaphase cells? At least a discussion would help.

This is a good suggestion and we have provided examples of the mitotic figures that we scored in Figure 2-supplement 1a. We now explain the meaning of the increase in prometaphase cells in the description of this result (lines 178-179).

Can the authors probe centrosomes (not by using γ-tubulin) and relate their presence or absence to p53 upregulation? This is an important point because a complete loss of centrosome is known to trigger p53 upregulation. This may be different in Haus6 KO. This could mean (i.e, centrosomes are normal in numbers or increase in numbers), p53 upregulation is regardless of centrosomes loss.

Indeed, we believe that p53 upregulation in *Haus6*-deficient brains is not caused by loss of centrosomes. Instead, our data suggest, as explained in the discussion, that mitotic delay caused by augmin deficiency is sufficient for p53 upregulation. We can now further support this conclusion by showing that centriole numbers are similar after *Haus6* cKO (Figure 2-supplement 1f, g).

I have a hard time to ascertain how the authors scored interphase cells that enriched with p53. Some representative images with identity markers will help.

Scoring p53-positive interphase cells is relatively straightforward since the p53 signal is nuclear and not observed in mitotic apical progenitors. We have included a magnified region of the tissue shown in Figure 2l, displaying PAX6/p53-positive nuclei of individual cells.

Looking at the p53 status in Haus6 KO animals, it is intriguing that p53 upregulation is not unique to centrosome loss. At this point, it becomes essential to thoroughly analyze the centrosome status to cross-check if Haus6 loss abrogates centrosomes; if so, how much.

Since centrosome number is linked to centriole number, we have addressed this point by quantifying centriole numbers by centrin staining in mitotic apical progenitors (see above) (Figure 2-supplement 1f, g).

Double KO could subside the cell death, but not tissue growth is impressive. So what is going on there? Is there a premature differentiation that leads to NPCs depletion? I believe the authors should generate 2D experiments with cells derived from these double KO animals compared to Haus6 KO and test if there is a premature differentiation that can lead to malformation of the forebrain. Here staining for the forebrain progenitor markers will additionally help (Perhaps FOXG1).

As already explained in more detail above, BrdU injection prior to fixation, followed by staining with cell cycle and cell type-specific markers did not reveal any evidence for premature differentiation after *Haus6* cKO.

Looking at Figure 6, it becomes clear that the double KOs have severe issues in maintaining the apical progenitors suggesting that they undergo premature differentiation before attaining a sufficient pool of NPCs. Testing this will bridge the paper between descriptive findings to mechanisms.

This point relates to the reviewer’s previous point: do *Haus6* cKO p53 KO apical progenitors prematurely differentiate? We believe that cell loss, tissue disruption, and aborted development may also be explained without premature differentiation. In the absence of p53, repeated abnormal mitoses (Figure 4, Figure 4-supplement 1), and the resulting increasingly severe chromosomal aberrations including DNA damage (Figure 5) may produce cells that eventually won’t be able to proliferate and function properly. This interpretation is also supported by the new BrdU injection experiments that did not detect evidence for premature differentiation.

The Discussion section is excellent, but it should add some human relevance. in particular, are there p53 dependent cell deaths that have been described in human tissues. In my opinion, it seems specific in the mouse brain. The discussion can also have statements about why the human brain is so sensitive even for mild mutations. I am not sure if those human mutations can cause similar defects in the mouse brain. Most of the mice based studies have been focusing on eliminating complete genes of interest.

We have included a section in the discussion to relate our findings to human brain development and the differences with results obtained in mouse models regarding the role of apoptosis (lines 421-424).

Significance:Overall, this is a very well done work but requires some more experiments for mechanisms understanding. Addressing those will make the paper fit to get published.

References:

Cunha-Ferreira I, Chazeau A, Buijs RR, Stucchi R, Will L, Pan X, Adolfs Y, van der Meer C, Wolthuis JC, Kahn OI, Schätzle P, Altelaar M, Pasterkamp RJ, Kapitein LC, Hoogenraad CC. 2018. The HAUS Complex Is a Key Regulator of Non-centrosomal Microtubule Organization during Neuronal Development. *Cell Reports* 24:791–800. doi:10.1016/j.celrep.2018.06.093

Goshima G, Mayer M, Zhang N, Stuurman N, Vale RD. 2008. Augmin: a protein complex required for centrosome-independent microtubule generation within the spindle. *The Journal of cell biology* 181:421–429. doi:10.1083/jcb.200711053

Lawo S, Bashkurov M, Mullin M, Ferreria MG, Kittler R, Habermann B, Tagliaferro A, Poser I, Hutchins JRA, Hegemann B, Pinchev D, Buchholz F, Peters J-M, Hyman AA, Gingras A-C, Pelletier L. 2009. HAUS, the 8-subunit human Augmin complex, regulates centrosome and spindle integrity. *Current biology : CB* 19:816–826. doi:10.1016/j.cub.2009.04.033

Uehara R, Nozawa R, Tomioka A, Petry S, Vale RD, Obuse C, Goshima G. 2009. The augmin complex plays a critical role in spindle microtubule generation for mitotic progression and cytokinesis in human cells. *Proceedings of the National Academy of Sciences of the United States of America* 106:6998–7003. doi:10.1073/pnas.0901587106

Watanabe S, Shioi G, Furuta Y, Goshima G. 2016. Intra-spindle Microtubule Assembly Regulates Clustering of Microtubule-Organizing Centers during Early Mouse Development. *Cell Reports* 15:54–60. doi:10.1016/j.celrep.2016.02.087

Zhu H, Coppinger JA, Jang C-Y, Yates JR III, Fang G. 2008. FAM29A promotes microtubule amplification via recruitment of the NEDD1–γ-tubulin complex to the mitotic spindle. *Journal of Cell Biology* 183:835–848. doi:10.1083/jcb.200807046